# TOKEN TO TOKEN LEARNING FROM VIDEOS

## ABSTRACT

We empirically study generative pre-training from videos. This paper does not describe a novel method, Instead, It studies a straightforward, yet must-know baseline given the recent progress of large language models pre-training for self-supervised vision pretraining. Our approach is conceptually simple and inspired by generative pre-training from text and images. To enable scaling to videos, we make several important improvements along the data, architecture, and evaluation axes. Our model, called *Toto*, is a causal transformer that generates videos autoregressively, one token at a time. We pre-train our model on a diverse set of videos with over 1 trillion visual tokens. Our tokens are quantized patch embeddings, and we use relative embeddings for coarse-to-fine pre-training. We conduct a large-scale study across a suite of diverse benchmarks, including image recognition, video classification, object tracking, robotic manipulation and scaling behaviours. We find that, despite minimal inductive biases, our approach achieves competitive performance across all benchmarks.

## 1 INTRODUCTION

In a paper published in 1951, Shannon, having just published the foundational papers of information theory, proposed a "guessing game" of *next word prediction* to estimate the entropy of English (Shannon, 1951). Nearly 70 years later, training a high-capacity transformer network (Vaswani et al., 2017) on this task, provided the generative pre-training backbone for Large Language Models (Radford et al., 2018; Devlin et al., 2019; Radford et al., 2019; Brown et al., 2020).

Less well known is the fact that in 1954, Fred Attneave (Attneave, 1954) proposed an analog of Shannon's task for images. To quote "We may divide the picture into arbitrarily small elements which we "transmit" to a subject (S) in a cumulative sequence, having them guess at the color of each successive element until they are correct. This method of analysis resembles the scanning process used in television and facsimile systems and accomplishes the like purpose of transforming two spatial dimensions into a single sequence in time".

While Attneave was concerned with images, in the context of 2024, we have to note the "Big Visual Data" is in videos, not images. While there are concerns that most of the text available on the Internet has already been used by the leading language models, in video we are barely started on the journey of Big Data exploitation.

As a step toward that goal, we propose a method for generative pre-training from videos. We build **Toto: Token to Token Video models** with necessary architectural changes to enable scaling to videos. Figure 1 shows our overall framework, we use multiple data sources, such as internet style exocentric videos, egocentric videos, and images to pre-train our models. Our pre-training objective is simple and follows language modeling, by predicting the next token. We then evaluate our models on various downstream tasks and show that generative video pre-training can lead to strong representations, for image, video understanding, and robot manipulation tasks.

In summary, this paper studies two central questions: (1) what is an appropriate architecture for generative pre-training in vision, and (2) what are the benefits of generative pre-trained features for vision tasks? First, we study the effects of various tokenization approaches (e.g. VQGAN (Esser et al., 2020), dVAE (Ramesh et al., 2021), patches (Dosovitskiy et al., 2020)) and find that, most of these perform similar to each other. We find that relative positional embeddings are better than absolute ones, allowing us to extend the context length and resolution in a straightforward manner. Our architecture enables us to train models on videos and images jointly. We pre-train three model

**Figure 1: Overall Framework.** Starting with images and video frames from a collection of datasets, we tokenize each frame/image into discrete visual tokens independently. We pre-train the transformer by predicting the next visual tokens, with a context length of 4K tokens of images or video frames. Once trained, we take the intermediate representations and evaluate them on various tasks.

sizes, up to 1 billion parameters on a large set of videos and images. We train all our models over 1 trillion tokens or the equivalent of 144 thousand hours of videos. We evaluate these models on various tasks such as image recognition, action recognition, object tracking, object permanence, and object manipulation with robots. Our findings show that, with minimal inductive biases our autoregressive generative pre-trained models perform competitively to approaches and show promising direction for training large-scale vision models on large quantities unfiltered video data. Finally, we study the scaling behaviours of *Toto* and show a power law relationship of loss vs optimal compute. We will release our models, training and evaluation code to enable further research on this direction.

## 2 RELATED WORK

Over the years self-supervised pre-training has proven to be effective in many areas including language, vision, and robotics. For vision, there are two main schools of thought, discriminative vs generative pre-training. Wu *et al.* (Wu et al., 2018) and SimCLR (Chen et al., 2020b) showed that instance discrimination training can learn strong discriminative features. Recently, MoCo (He et al., 2020) and DINO (Caron et al., 2021) show the effectiveness of strong visual representations on various downstream tasks. Generative pre-training on the other hand, learns to model the data distribution. In language models, generative pre-training has become the de facto standard for training large models. On the vision models, we are still exploring the powers of generative models.

**Masked Modeling:** BEiT (Bao et al., 2021) and MAE (He et al., 2022) follows the BERT (Devlin et al., 2018) style masked modeling of images. Compared to BERT, MAE uses asymmetric encoder-decoders, allowing it to be very efficient at training with high masking ratios. ST-MAE (Feichtenhofer et al., 2022), VideoMAE (Wang et al., 2023a) apply this masked modeling to videos, by masking a large amount of tokens during pre-training and predict the masked tokens with a light-weight decoder.

**Autoregressive Modeling:** For Autoregressive pre-training, PixelCNN (Van den Oord et al., 2016) and PixelRNN (Van Den Oord et al., 2016) proposed generating pixels one by one using convolution and bidirectional LSTMs. With the introduction of the transformers (Vaswani et al., 2017), Image-Transformers (Parmar et al., 2018) showed generating pixels with causal local attention performs better than previous CNN and RNN-based methods. While all of these methods focused on the generation quality of the pixels, iGPT (Chen et al., 2020a) showed that generative pre-training is also a good way to learn strong visual representations for recognition tasks. AIM (El-Nouby et al., 2024) on the other hand uses patch embedding rather than any pre-trained models for tokenization, however, it trains on Data Filtering Networks (Fang et al., 2023) with clip filtered data. Compared to these works, we do not use any supervision during our pre-training and utilizes image and videos jointly. VisionMamba (Zhu et al., 2024) also showed how to utilize sequence models with bidirectional state-space modeling for supervised vision tasks.

**Robot Manipulation:** Control of action, based on pixel observations gives a good signal on how good the learned representations are at estimating the state of the object being manipulated. MVP (Xiao et al., 2022) showed that manipulation tasks can be learned with better sample efficiency when using pre-trained vision models for encoding pixel observations, and it also generalizes to real-world tasks (Radosavovic et al., 2022). For video based generative pretraining (Wu et al., 2023) showed the benefits of video-language pretrained models for learning robot policies.

## 3 APPROACH

We train a casual transformer model to predict the next patch-tokens in images and videos. This is equivalent to the next token prediction in large language models. From the vast collection of images and videos, every patch is tokenized into a discrete token, and the transformer is trained to predict the next token, using raster scan ordering. We pre-train our models on over one trillion tokens. Finally, we evaluate the learned representations of these models on various downstream tasks on image classification, action classification, action anticipation, video tracking, object permanence, robotic manipulation tasks and scaling behaviours.

### 3.1 PRE-TRAINING

Given a large collection of images and videos, we tokenize all of them into a 1D sequence using raster scan ordering. This produces a dataset of tokens, $\{x_1^j, x_2^j, x_3^j, ..., x_n^j\}$ where $j$ is the sample either from a video or an image and $n$ is the number of tokens in an image or a video. We model the density $p(x)$ as :

$$p(x^j) = \prod_{i=1}^{n} p(x_i^j | x_{i-1}^j, x_{i-2}^j, ..., x_1^j, \theta) \tag{1}$$

Here, $\theta$ is the model parameters, which can be optimized by minimizing the negative log-likelihood loss:

$$\mathcal{L}_{\text{pre-train}} = \mathbb{E}_{x^j \sim X} - \log p(x^j). \tag{2}$$

Using this loss, we pre-train our models at different sizes on over one visual trillion tokens. These tokens are generated from images and video. Figure 2 shows the training loss of 3 differently sized models with 120m, 280m and 1.1b parameters.

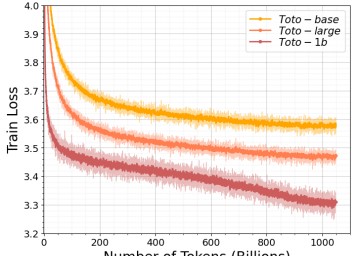

**Figure 2: Training Loss Curves:** We show the training loss curves for base, large, and 1b models trained with tokens from dVAE (Ramesh et al., 2021) with a vocabulary size of 8k and context length of 4k tokens (equivalent to 16 images or video frames).

### 3.2 ARCHITECTURE

Our model is a transformer model (Vaswani et al., 2017) with causal attention. We apply recent advancements in language modeling such as pre-norm using RMSNorm (Zhang & Sennrich, 2019), SwiGLU activation (Shazeer, 2020), and RoPE for positional embeddings (Su et al., 2024), following LLaMa (Touvron et al., 2023). We train the autoregressive transformer model for the next token prediction task, at different scales (base, large and 1b models). For more architecture details see Table 1. We train all these models with a batch size of 1 million tokens. We use AdamW (Loshchilov & Hutter, 2017) with a maxiumum learning rate of $3e{-}4$, and $\beta_1 = 0.9, \beta_2 = 0.95$. We decay the learning rate with a cosine schedule, after 2000 steps of warmup (Touvron et al., 2023).

| Model | # params | dimension | # of heads | # of layers |
|-------|----------|-----------|------------|-------------|
| base  | 120m     | 768       | 12         | 12          |
| large | 280m     | 1024      | 16         | 16          |
| 1b    | 1.1b     | 2048      | 16         | 22          |

**Table 1: Model Architecture**: We pre-train models at different scales, only on visual tokens from images and videos. All of these models use relative positional embeddings RoPE (Su et al., 2024).

### 3.3 DATASET

We train our models on a mixture of various datasets. Table 2 shows the total number of images and videos used for training data, the total number of tokens, as well as the number of hours of videos in each dataset. Together these datasets contain over 100,000 hours of video data and about 2.5 trillion visual tokens. During training, each batch is sampled at different ratios of datasets. Each batch approximately contains 20% of ImageNet images, 10% of Ego4D videos, 10% of Kinetics videos, and 60% of HowTo100m videos. Our full training only utilized about 1 trillion tokens.

| Datasets | # of instances | # of tokens | # of hours |
|----------|---------------:|------------:|-----------:|
| ImageNet | 13.9 m | 3.6 b | - |
| Kinetics-600 | 0.53 m | 41.3 b | 1496 |
| Ego4D | 52.1 k | 103 b | 3750 |
| HowTo100m | 1.172 m | 2560 b | 92,627 |

**Table 2: Pre-training Datasets**: We use both image datasets (Imagenet (Russakovsky et al., 2015)) and video datasets (Kinetics600 (Carreira et al., 2019), Ego4D (Grauman et al., 2022), HowTo100m (Miech et al., 2019)) with different mixing ratios during the pre-training of our models. The whole training data contains over 100,000 hours of videos and up to 2.5 trillion visual tokens.

We use dVAE tokenizer with a vocabulary of 8k tokens, from Dall-E (Ramesh et al., 2021) as our tokenizer. Using an image-based tokenizer allows training on both images and videos and testing on respective downstream tasks. While VQGAN (Esser et al., 2020) tokenizers provide sharper images, these models were trained with perceptual loss (Larsen et al., 2016; Johnson et al., 2016), thus indirectly ingesting VGG-net (Simonyan & Zisserman, 2014) ImageNet label information.

All raw pixel frames or images are tokenized into 256 discrete tokens. We take a video and resize it such that its shortest size is $R$ pixels, and then take a random crop of $R \times R \times T$, and sample every 4 frames where $T$ is the number of frames. We use dVAE (Ramesh et al., 2021) with the vocabulary of 8k vocabulary to tokenize every frame independently. For dVAE we set $R = 128$, to get $16 \times 16$ discrete tokens. Once every frame is mapped into a set of discrete tokens we will have $T \times 256$ tokens per each video. We pre-train all the models with $T = 16$, thus all the models were per-trained for a context length of 4096 tokens.

When training with images and videos, 16 video frames are sampled to create 4k tokens. For images, we randomly sample 16 images and create a sequence of 16 image frames to generate 4k tokens. Finally, we add start and end tokens for each sequence, for videos we use [1] as the start token, and for images we use [3] as the start token, and all sequences have an end token of [2].

### 3.4 DOWNSTREAM TRANSFER

The idea of large pre-trained models is that they were trained at a large compute scale, and then these models can be easily used for various downstream tasks without requiring task-specific design or lots of computing for transfer. The learned representations are general enough to transfer to various tasks.

Our transformer architecture is a decoder-only model, with a sequence of self-attention layers and MLP layers. Let's assume $H^l$ is the intermediate representations after layer $l$, then the tokens at layer $l + 1$ are computed as follows:

$$\widehat{H}^{l+1} = \texttt{layer-norm}(H^l) \tag{3}$$

$$\widehat{H}^{l+1} = \widehat{H}^{l+1} + \texttt{MHSA}(\widehat{H}^{l+1}) \tag{4}$$

$$H^{l+1} = \widehat{H}^{l+1} + \texttt{SwiGLU}(\texttt{MLP}(\widehat{H}^{l+1})) \tag{5}$$

Here, MHSA is multi-head self-attention (Vaswani et al., 2017) and we use SwiGLU activation (Shazeer, 2020) in the MLPs.

Let's say the representations $H^l = \{h_1, h_2, ..., h_t\} \in \mathcal{R}^{t \times d}$ at layer $l$, has $t$ number of tokens with hidden dimension $d$. For linear probing the model, we take these tokens and, and apply global average pooling (Lin et al., 2013) over $t$ tokens to get the intermediate representation $\tilde{H}^l = \frac{1}{t}\sum_t h_t$. Then we train a linear layer on top of this representation on the downstream task.

MAE (He et al., 2022) or BEiT (Bao et al., 2021) have a uniform structure when it comes to which token attends which tokens, however in language modeling later tokens attend more tokens than the tokens at the beginning. Due to this skewed nature equally weighting all the tokens affects the downstream performance. Attention pooling allows to dynamically weight the tokens, ideally giving more weight to tokens that see more tokens. This requires learning $W_k$ and $W_v$ matrices and a query token $q$. The query token cross-attends the intermediate tokens and combines them into a single vector. Then we learn a linear layer on top of this representation for downstream tasks. While this whole function is not linear anymore, we argue that for a casual model equally averaging the tokens is not fair. This has shown to be effective in recent works as well (El-Nouby et al., 2024).

## 4    EXPERIMENTS

We evaluate our pre-trained models on various downstream tasks such as ImageNet classification, Kinetics action recognition, Ego4D action anticipation, Semi-Supervised tracking, and Robotic manipulation tasks. First, we discuss various design choices for pre-training and evaluation strategies for our method. All the models for studying the design choices are `large` models trained for 400 epochs on the ImageNet-1k dataset.

### 4.1    DESIGN CHOICES

**Tokenizer:**   The are various options available for tokenizing an image or a video. We could use discrete tokenizers such as dVAE, and VQGAN, or simple patch-based continuous tokenization. To study the behaviour of various tokenizers we pre-train a large model on ImageNet for 400 epochs. Using linear probing at an optimal intermediate layer, we evaluate different models on ImageNet classification tasks.

Table 3 shows linear probing accuracy when trained with various tokenizers. VQGAN (Esser et al., 2020) and dVAE (Ramesh et al., 2021) perform similarly with the same resolutions. However, VQGAN is contaminated with ImageNet label information via perceptual loss. In addition to that, as shown in Figure 3, dVAE tokens have full coverage compared to VQGAN tokens on their 1-gram distributions. Please see in the supplementary material for more details. Regressing normalized-patch targets from patch embeddings performs slightly worse than classifying discrete tokens as targets. Additionally, discrete tokens as targets and patch embeddings as inputs perform poorly compared to other methods at the given input-output resolutions. Overall, Table 3 shows that various ways of tokenization have *little effect* on ImageNet linear probing accuracy.

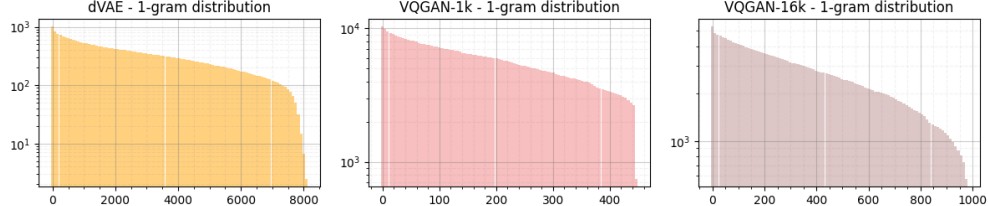

**Figure 3: 1-gram Distribution of Various Tokens:** This Figure shows the distribution of 1-gram tokens of various tokenizers (dVAE (Ramesh et al., 2021), VQGAN-1k, VQGAN-16k (Esser et al., 2020)) on Imagenet validation set. Note that, dVAE has almost full convergence of the tokens while VQGAN has less than 50% coverage of the tokens.

| Input-Target | # tokens | Vocabulary | Top1 |
|---|---|---|---|
| VQGAN-VQGAN | 16x16 | 16k | 61.3 |
| VQGAN-VQGAN | 16x16 | 1k | 61.1 |
| dVAE-dVAE | 32x32 | 8k | 61.2 |
| dVAE-dVAE | 16x16 | 8k | 53.2 |
| patch-patch | 16x16 | - | 60.6 |
| patch-dVAE | 16x16 | 8k | 58.5 |

**Table 3: ImageNet Linear Probing Accuracy with Various Tokenizers:** We compare discrete (dVAE, VQGAN) and patch embedding as input and target for pre-training our models. ImageNet top-1 accuracies are computed by linear probing at the 9th layer of the `large` model.

**How to probe:**   As discussed in Section 3.4 we probe the pre-trained models at the same layer with attention pooling and average pooling, followed by linear layer. Table 5 shows attention pooling performs 7.9% higher than average pooling on the ImageNet classification task. For attention pooling, we keep the embedding dimension the same as the intermediate feature dimensions.

**Resolution:**   When training with dVAE tokens, a 256x256 image results in 1024 tokens, this is four times more number of tokens compared to patch embeddings or VQGAN tokens. If we reduce the

| Method | Compute | Top1 |
|---|---|---|
| dVAE/16 | $1.42 \times 10^{17}$ | 53.2 |
| dVAE/32 | $5.68 \times 10^{17}$ | 61.2 |
| dVAE/16→32 | $2.13 \times 10^{17}$ | 63.2 |
| dVAE/16→32[†] | $2.13 \times 10^{17}$ | 64.4 |

| Method | tokens | pooling | Top1 |
|---|---|---|---|
| dVAE | 16x16 | Average | 53.2 |
| dVAE | 16x16 | Attention | 61.1 |

**Table 4: Token Resolution:** While the performance is lower for a low-resolution model, when finetuned for next-patch prediction at a higher resolution, its performance surpasses the full-resolution pre-trained model.

**Table 5: Attention vs Average Pooling:** When probed at the same layers, attention pooling performs much better than average pooling of intermediate tokens.

number of tokens to 256, then the effective image resolution becomes 128x128. Table 4 shows a clear drop in performance when pre-training the model at 128x128 resolution. However, due to the use of relative positional embeddings (RoPE (Su et al., 2024)), we can easily finetune the 128x128 (or 16x16 token equivalent) model for higher resolution. Surprisingly, this does better than pre-training at 256x256 resolution and requires only one epoch of finetuning. Not only does this improve the performance, but the pre-training also becomes cheaper compared to full-resolution pre-training.

**Architecture:** We train various language models from GPT2 (Radford et al., 2019) with absolute sine-cosine positional embeddings, and non-transformer based model Mamba (Gu & Dao, 2023) only using dVAE tokens. We mimicked the GPT2 architecture and do architecture comparisons. We compare these models with *Toto*. We evaluate linear probing performance at each layer of these models and report the best performance in Table 6.

| Model | #$\theta$ | Top1 |
|---|---|---|
| GPT2 (Radford et al., 2019) | 280 m | 48.5 |
| Mamba (Gu & Dao, 2023) | 290 m | 40.7 |
| *Toto* | 280 m | 53.2 |

**Table 6: Architecture:** We compare similar models GPT2 (Radford et al., 2019), and non-transformer models, Mamba (Gu & Dao, 2023) with *Toto*. *Toto* performs best on ImageNet linear probing task.

**Probing Layer:** When probing the pre-trained models, especially the decoder-only model best performance is observed at the middle layers. This behavior is first observed in iGPT (Chen et al., 2020a). Figure 4 shows the peak performance on recognition occurs at about 50% of the depth of the model. This behavior holds across all model sizes. While in MAE (He et al., 2022) and BEiT (Bao et al., 2021) encoder-decoder models, due to the uneven nature of the encoder and decoder, the best features are observed at the top of the encoder layers. However, on decoder-only models with uniformly distributed layers, the last layers perform worse on recognition tasks, mainly because these layers are trained to reconstruct the input. More probing results with various tokenizers, resolutions, and probing methods are shown in the supplementary material.

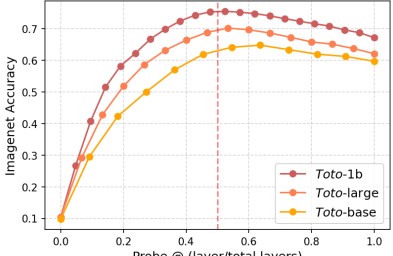

**Figure 4: Probing at Different Layers:** We show the attention-probing performance at each layer of our three models. Peak performance is observed at around 50% depth of the models.

## 4.2 IMAGE RECOGNITION

To measure the representation quality of our pre-trained models, we evaluate our models on ImageNet-1k (Deng et al., 2009) classification. We apply a probe at each layer of the model, with attention pooling, and choose the optimal layer with the highest classification accuracy. We fine-tune the pre-trained models further by applying self-supervised loss, together with cross-entropy loss applied for probing layers (with stop-gradients). We train the probing layers for 90 epochs, with a learning rate of $6e^{-5}$. We also use layer decay of 0.9 to reduce the learning rate at the early layers of the model. During this stage, all the models are fine tuned with $32 \times 32$ token resolution, on the self-supervised

---

[†]Fine-tuning with higher base values of the RoPE embeddings (50,000) leads to better accuracy.

loss, and increase the base value of the RoPE (Su et al., 2024) embeddings from 10,000 to 50,000 support larger resolution.

Table 7 shows the ImageNet top-1 accuracy of our `base, large` and `1b` models. First, there is a clear difference in terms of classification performance when it comes to discriminative models vs generative models. Instance discriminative models such as SimCLR (Chen et al., 2020b), and DINO (Caron et al., 2021) are trained to separate samples from each other and they are designed to perform well on discriminative tasks. On the other hand, generative models are *just* trying to model the data distribution. While achieving comparable performance to other generative models on image recognition, among autoregressive generative models, our model achieved the highest top-1 accuracy. The scaling of data, and the use of tokens instead of pixels, allows our one billion parameter model to achieve similar performance compared to iGPT (Chen et al., 2020a) 7 billion models.

| Method | Arch | #$\theta$ | Top1 |
|---|---|---|---|
| *Discriminative Approaches* | | | |
| SimCLR (Chen et al., 2020b)† | RN50x2 | 94 | 74.2 |
| BYOL (Grill et al., 2020)† | RN50x2 | 94 | 77.4 |
| SwAV (Caron et al., 2020)† | RN50x2 | 94 | 73.5 |
| DINO (Caron et al., 2021) | ViT-B/8 | 86 | 80.1 |
| DINOv2 (Oquab et al., 2023) | ViT-g/14 | 1011 | 86.4 |
| *Generative Approaches* | | | |
| AIM (El-Nouby et al., 2024) | ViT-3B/14 | 3B | 82.2 |
| BEiT-L (Bao et al., 2021) | ViT-L/14 | 307 | 62.2 |
| MAE (He et al., 2022) | ViT-L/14 | 307 | 80.9 |
| iGPT-L (Chen et al., 2020a)† | GPT-2 | 1386 | 65.2 |
| iGPT-XL (Chen et al., 2020a)† | GPT-2 | 6801 | 72.0 |
| *Toto*-base | LLaMA | 120 | 64.7 |
| *Toto*-large | LLaMA | 280 | 71.1 |
| *Toto*-1b | LLaMA | 1100 | 75.3 |

**Table 7: ImageNet Results:** We compare discriminative and generative models on ImageNet (Deng et al., 2009) recognition task. While achieving comparable performance among generative models, our models model achieves the highest accuracy on autoregressive modeling. †models are evaluated with linear probing.

### 4.3 ACTION RECOGNITION

We use Kinetics-400 (K400) (Kay et al., 2017) for evaluating our models on action recognition tasks. Similar to ImageNet evaluation, we apply a probe at each layer of the model, with attention pooling, and choose the optimal layer with the highest action classification accuracy. We also fine-tune the pre-trained models on a self-supervised next-patch prediction task while training the probing layers with a classification loss. All our video models are trained with 16 frames, thus with a context length of 4096 tokens per video. When evaluating videos, we follow the protocol in SlowFast (Feichtenhofer et al., 2019). Unlike ImageNet where we evaluate the models at 256x256 resolution, on videos we only evaluate our models at 128x128 resolution, to keep the number of tokens in a similar budget.

Table 8 shows the Kinetics-400 top-1 accuracy of our `base, large` and `1b` models. Similar to ImageNet results in Table 7, we see that discriminately trained models perform better than generative models. Our models achieve comparable performance among generative models, and first to show competitive performance on action recognition with autoregressive generative modeling. All the models are trained and evaluated with 16 frames with a stride of 4 frames.

### 4.4 ACTION FORECASTING

While the Kinetics dataset captures internet-style exocentric videos, Ego4D (Grauman et al., 2022) videos capture day-to-day life egocentric videos. A general vision model should be able to reason about both exo and ego-centric videos. Task-wise, Kinetics requires the model to reason about the

| Method | Arch | Top1 |
|--------|------|------|
| *Discriminative Approaches* | | |
| I-JEPA (Assran et al., 2023) | ViT-H/16 | 74.5 |
| OpenCLIP (Cherti et al., 2023) | ViT-G/14 | 83.3 |
| DINOv2 (Oquab et al., 2023) | ViT-g/14 | 84.4 |
| InternVideo (Wang et al., 2022) | - | 73.7 |
| VATT (Akbari et al., 2021) | - | 75.1 |
| *Generative Approaches* | | |
| Hiera (Ryali et al., 2023) | Hiera-H/14 | 77.0 |
| MVD (Wang et al., 2023b) | ViT-H/14 | 79.4 |
| VideoMAE (Wang et al., 2023a) | ViT-L/14 | 79.8 |
| *Toto*-base | LLaMA | 59.3 |
| *Toto*-large | LLaMA | 65.3 |
| *Toto*-1b | LLaMA | 74.4 |

**Table 8: K400 Results:** We compare discriminative and generative models on Kinetics-400 (Kay et al., 2017) action recognition task. While achieving comparable performance among generative models, our models are the first to show the competitive performance on K400 with autoregressive pre-training, and shows scaling nature with large model sizes.

action using full context (e.g. the model has seen the action), while the Ego4D short-term action anticipation v1 task requires models to predict future actions from past context. We use our models as the backbone for the pyramid network used in StillFast (Ragusa et al., 2023) extract tokens at 5 layers and fuse them with the pyramid network. We fully fine-tuned our model with self-supervised next-patch loss along with task-related losses, and we observed having self-supervision loss improves overall performance. Table 9 shows the performance of our `large` model on the Ego4D short-term action anticipation task. This task requires predicting the object to be interacted with (noun) and the type of interaction (verb) as well as time to contact (ttc) from the last seen frame to an estimated time between object-hand contact. As shown in Table 9, these tasks are difficult with maximum overall mean-average precision of 2.70.

| Method | Noun | N+V | N+TTC | Overall |
|--------|------|-----|-------|---------|
| FRCNN+Rnd (Grauman et al., 2022) | 17.55 | 1.56 | 3.21 | 0.34 |
| FRCNN+SF (Grauman et al., 2022) | 17.55 | 5.19 | 5.37 | 2.07 |
| Hiera-large (Ryali et al., 2023) | 14.05 | 6.03 | 4.53 | 2.12 |
| StillFast (Ragusa et al., 2023) | 16.20 | 7.47 | 4.94 | 2.48 |
| VideoMAE-large (Wang et al., 2023a) | 15.16 | 6.72 | 5.26 | 2.55 |
| MAE-ST-large (Feichtenhofer et al., 2022) | 13.71 | 6.63 | 4.94 | 2.60 |
| *Toto*-large | 15.20 | 6.75 | 5.41 | 2.70 |

**Table 9: Ego4D Results:** Our model achieves comparable mean-average precision compared to previous work. We compare our method with, FRCNN+Rnd (Grauman et al., 2022), FRCNN+SF (Grauman et al., 2022), Hiera (Ryali et al., 2023), StillFast (Ragusa et al., 2023), VideoMAE (Wang et al., 2023a), and MAE-ST (Feichtenhofer et al., 2022).

## 4.5 VIDEO TRACKING

In this section, we study our pre-trained models on label propagation using the protocols in (Jabri et al., 2020). Compared to previous tasks such as classification, and forecasting, this evaluation requires zero adaptation of the features. We use the features from the last $n$ frames to find the nearest neighbor patch in the current frame, and then propagate the segmentation masks from the previous frames to the current frame and this requires no fine-tuning. Comparison with Dino (Caron et al., 2021) and MAE (He et al., 2022) is show in Table 10 and qualitative results are shown in Figure 5.

| Method (Res/Patch) | J&F | J | F |
|--------------------|-----|-----|-----|
| DINO-base (224/8) | 54.3 | 52.5 | 56.1 |
| DINO-base (224/16) | 33.1 | 36.2 | 30.1 |
| MAE-base (224/16) | 31.5 | 34.1 | 28.9 |
| *Toto*-base (256/8) | 42.0 | 41.2 | 43.1 |
| *Toto*-large (256/8) | 44.8 | 44.4 | 45.1 |
| *Toto*-1b (256/8) | 46.1 | 45.8 | 46.4 |
| *Toto*-large (512/8) | 62.4 | 59.2 | 65.6 |

**Table 10: DAVIS Tracking:** We report J, F, and J&F scores at the peak layers of each model. We achieves comparable performance as DINO and at large resolution (512), it outperforms all methods.

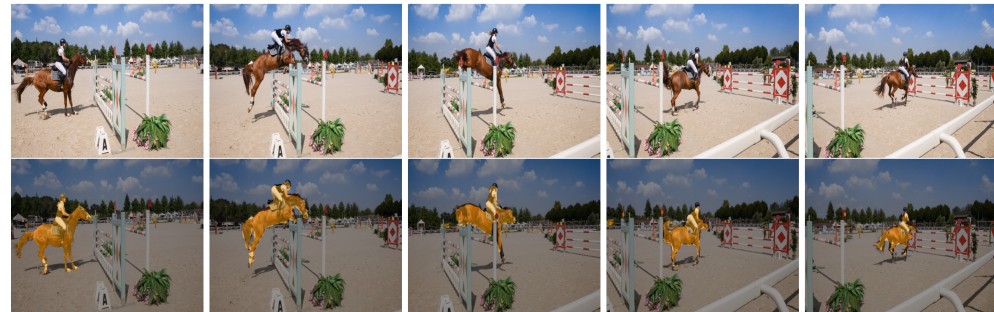

**Figure 5: Semi Supervised Tracking:** We follow the protocol in STC (Jabri et al., 2020), start with the GT segmentation mask, and propagate the labels using the features computed by *Toto*-large. The mask was propagated up to 60 frames without losing much information.

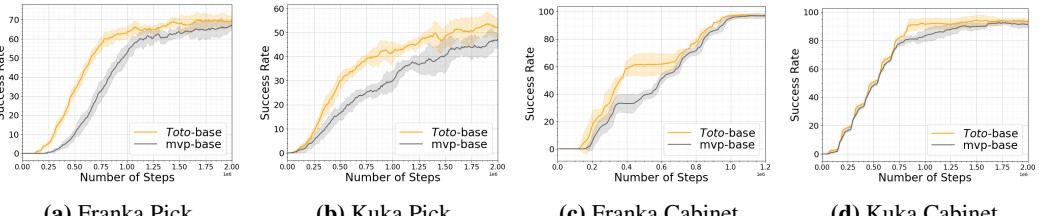

| **(a)** Franka Pick | **(b)** Kuka Pick | **(c)** Franka Cabinet | **(d)** Kuka Cabinet |

**Figure 6: Robot Manipulation Results:** We compare MAE-base (Xiao et al., 2022) with our `base` pre-trained model on robot manipulation tasks. We evaluate each model based on the mean success rate over training steps. Our model was able to learn these tasks faster than MAE model, across two robots and two tasks.

## 4.6 ROBOTICS

In this section, we study the effectiveness of our pre-trained representations for robotic manipulation. We consider tasks in both simulation and in the real world. Real world experiments needs to run at real time, there for we only use *Toto*-base models, in both setting. Despite being a small model, *Toto*-base can achieve better performance in simulation and on-par performance to state-of-the-art robot models in real world experiments.

**Simulation experiments:** Following the protocols in MVP (Xiao et al., 2022), we use our visual pre-trained models to embed pixel observations. The model is frozen and we only take tokens at an intermediate layer, apply average pooling, and learn the linear layer on top to embed pixel observations. These observations are used to train DAgger policies for 4 different tasks: Franka-pick 6a, Kuka-pick 6b, Franka-cabinet 6c, and Kuka-cabinet tasks 6d. Figure 6 shows the mean success rate over training steps. Compared to the MVP baseline, our model was able to learn these tasks faster with better sample efficiency across robots and tasks. For fair comparisons, we use the best MAE model from MVP (Radosavovic et al., 2022) which is trained on ImageNet (Deng et al., 2009), Ego4D (Grauman et al., 2022) and 100DOH (Shan et al., 2020) datasets.

**Real-world experiments:** Next, we evaluate our pre-trained representations in the real world. We follow the setup from (Radosavovic et al., 2022). We extract vision features using a pre-trained vision encoder and train a controller on top of frozen representations using behavior cloning. Specifically, we consider a cube picking tasks using a 7 DoF Franka robot, shown in Figure 7. We use the demonstrations provided by (Radosavovic et al., 2023). In Table 11 we compare our model to a vision encoder from (Radosavovic et al., 2022). We report the success rate over 16 trials with variations in object position and orientation. Our model performs favorably to a vision encoder pre-trained for robotics.

| Model | # traj | Success |
|---|---|---|
| MVP | 240 | 75% |
| *Toto*-base | 240 | 63% |

**Table 11: Real-world Experiments:** We compare MVP (Radosavovic et al., 2022) and *Toto* on a Franka cube-picking task in the real world. Features from both models are pre-trained, frozen, and passed into a learning module trained with behavior cloning using the same demonstrations. We see that our approach performs comparably to the state-of-the-art vision backbone for robotics, despite not being designed with the robotic application in mind.

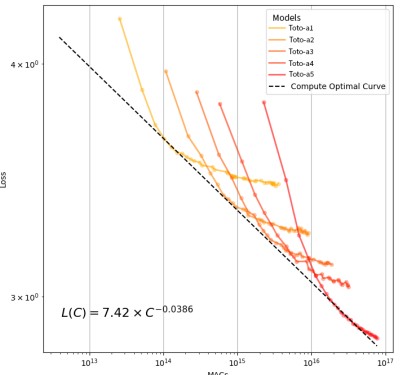

**Figure 7: Real-world Deployment:** We show an example episode of our policy performing the cube picking task on a Franka robot in the real world.

### 4.7 OBJECT PERMANENCE

To quantitatively measure the performance of how well the model understands object permanence, we evaluate our models on CATER localization task (Girdhar & Ramanan, 2019). Here, a ball is moving in the scene, and the task is to find its location in the 6 by 6 grid. We finetune our model on this task at temporal resolutions 16, and 32 frames. In all resolutions, our pre-trained models were better at localizing the target object compared to models trained specifically for this task. Table 12 shows the performance on the CATER snitch localization task.

| Method | Model | 16 | 32 |
|--------|-------|------|------|
| V3D | ResNet | 55.2 | 69.7 |
| TFC V3D | ResNet | 54.6 | 70.2 |
| *Toto*-large | LLaMa | 62.8 | 72.9 |

**Table 12: Object Permanence:** CATER (Girdhar & Ramanan, 2019) object localization task, where the object is hidden under or obstructed by other objects. The model is trained to predict its coarse location. Our model performs better than previous methods on this task at 16 and 32 temporal resolutions.

### 4.8 COMPUTE OPTIMAL SCALING

We study the scaling behaviours of *Toto* using $\mu$-Parameterization (Yang et al., 2022). First we train various models a1-a6, with linearly increasing hidden size and number of layers (Table 15), and we used VQGAN tokenizer (Esser et al., 2020). Then we tune the learning rate for these models, with $\mu$-Parameterization (Yang et al., 2022). Figure 16 shows optimal learning rate of $2^{-7}$ for all the model widths. Once we find the optimal learning rate, we train a1-a6 models on our data mixture, as mentioned in Table 2. Figure 8 shows the loss vs compute of *Toto* models. This shows a clear power law relationship with compute and validation loss. Based on these experiments *Toto* shows a power law of $L(C) = 7.42 \cdot C^{-0.0386}$. Interestingly, if we look at GPT3 (Brown, 2020) power law relationship, it has $L(C) = 2.57 \cdot C^{-0.048}$. While these are not comparable directly, but the scaling coefficient shows how much change in loss for an added extra compute. This shows, that visual next token models scales, but at a slower rate than language only models.

**Figure 8: Scaling *Toto*:** We train multiple variants of *Toto*, with increasing hidden size and depth, with optimal learning rates. We plot the validation loss vs the compute spent on training in MACs. This shows a clear scaling behaviour with optimal compute.

## 5 CONCLUSION

We present an approach *Toto*, for generative pre-training from videos. We build on prior work on generative pre-training from images and make architectural improvements to enable scaling to videos, including the use of quantized patch embeddings, relative position information. We collect a large video dataset and conduct a large-scale empirical study across a range of diverse tasks, including image recognition, video classification, object tracking, trajectory prediction, and robotic manipulation. We perform extensive ablation studies to understand different design choices and compare our approach to strong baselines across different tasks. We find that, despite minimal inductive biases, our approach achieves competitive performance across all tasks. Finally, we studied the scaling behaviours of visual next token prediction models, and showed it scales with compute, but at a slower rate than text based next token prediction models.

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
