## A APPENDIX

### A.1 ADDITIONAL DISCUSSION

The next patch prediction for visual pre-training is equivalent to the next token prediction in large language models. However, most languages have a clear sequential nature, therefore there is a clear definition for the next word. This also makes the next word prediction task relatively harder, since the model requires learning to extrapolate the data. On the other hand, images and videos, especially over the spatial dimensions lack a sequential nature. We follow the previous works (Chen et al., 2020a; Van Den Oord et al., 2016) to make the images and videos into a 1D sequence by scanning the patches in raster order. While this ordering allows for example to learn to predict the bottom half of the image from the top part of the image, in many places, the tokens can be predicted by interpolating rather than extrapolating. On the time axis, yes, there is a clear sequential nature, however, video frames compared to text tokens are more redundant, making the next frame prediction task much easier.

### A.2 LIMITATIONS

In this work, we introduced *Toto*, for generative pre-training from videos, showing its strong performance across a variety of visual tasks. Despite its competitive performance, this approach has limitations. A significant limitation stems from the use of internet videos, which, unlike carefully curated datasets, introduces challenges related to data quality and diversity. This variance in data quality can impact model performance, especially when compared to models trained on more curated datasets. Another challenge is the considerable computational demand of our large-scale pre-training approach, potentially placing it beyond the reach of researchers with limited computational resources. Furthermore, we have not yet fully assessed our method's effectiveness in dealing with dense prediction tasks, fine-grained recognition, or comprehending complex temporal dynamics over extended time frames. These areas represent key opportunities for further research, aiming to broaden the fruitfulness of generative pre-trained models.

### A.3 PREFIX ATTENTION

During fine-tuning, we experimented with causal and full attention. On ImageNet, our base model achieved full attn: 82.6% vs causal attn: 82.2%. Even though our models are *not pre-trained with prefix attention*, still able to utilize full attn at fine-tuning. This is an unrealized benefit of training with videos, (a middle token in say, 8th frame won't see the rest half of the 8th frame, but have seen all the tokens from 7th frame, which are similar because of video, hence approximating full attention at pre-training)

### A.4 FULL FINE-TUNING

We fine-tuned our models on ImageNet, and performance is close to SOTA, compared to linear probing (where we only use causal attention). But during the fine-tuning, we use full attention. We will add this comparison including larger models variants to the main paper.

| DINO | MoCo v3 | BEiT | MAE | *Toto* |
|------|---------|------|-----|--------|
| 82.8 | 83.2 | 83.2 | 83.6 | 82.6 |

**Table 13: Full Fine Tuning Performance:** Comparison of different methods and their full fine tuning performance on ImageNet-1K.

### A.5 IGPT VS *Toto* ON IMAGENET

Table 7 shows ImageNet evaluation performance. However, iGPT (Chen et al., 2020a) models are evaluated only using linear probing. To have a fair comparison, between iGPT and *Toto*, we reevaluated our models using linear probing. Both models have causal attention and are trained on auto-regressive objectives. On the same model sizes, about 1 billion parameters, our achieve 66.2% while the similar iGPT model's ImageNet performance is 65.2%. This fair evaluation suggests the modifications made on *Toto* have clear benefits over iGPT.

| Method | Arch | #$\theta$ | Top1 |
|---|---|---|---|
| iGPT-L (Chen et al., 2020a) | GPT-2 | 1386 | 65.2 |
| *Toto*-1b | LLaMA | 1100 | 66.2 |

**Table 14: ImageNet Linear Probing Results:** *Toto* performs better than similar size iGPT models.

## A.6 PROBING ACROSS LAYERS

As shown in Figure 4 for the ImageNet classification task, different layers for the model contribute to the task differently for the image classification task. To study this behavior across multiple tasks, we train probing layers for all other tasks such as action recognition, object tracking, and robot manipulation. Figure 9 shows probing performance across layers, model size, and tasks. It shows that action recognition follows a similar trend to ImageNet classification tasks, having peak performance at the middle of the model stacks.

While Object tracking also shares a similar trend with image classification and action recognition, object manipulation shows an interesting trend of the last layers performing well as middle layers from picking objects. Compared to the first three tasks, robot manipulation has a generative nature as a task and can benefit from generative pre-training.

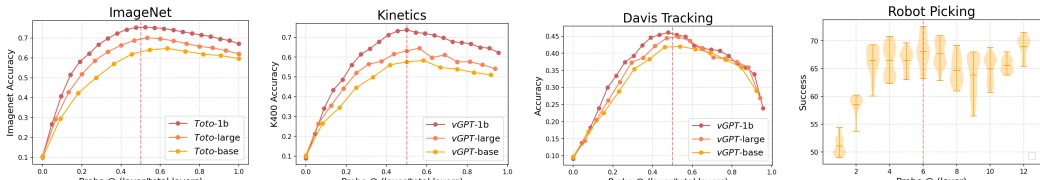

**Figure 9: Probing Across Layers, Models, and Tasks:** We study the behavior of our models across multiple layers and tasks. For image classification, action recognition, and object tracking, all the models behave similarly and peak around 50% of the model depth. This behavior is observed across all model sizes. Robot tasks show a different behaviour, where the middle layers perform good at picking the object, last layers also perform good as middle layers.

## A.7 $\mu$-PARAMETERIZATION

To study the scaling behaviours of *Toto* using $\mu$-Parameterization (Yang et al., 2022). First we train various models a1-a6 (in Table 15), with hidden sizes (64-1536) and number of layers (12-48), increasing linearly and we used VQGAN tokenizer (Esser et al., 2020). Then we tune the learning rate for these models, with fixed depth using $\mu$-Parameterization (Yang et al., 2022). Figure 16 shows optimal learning rate of $2^{-7}$ for all the model widths. Once we find the optimal learning rate, we train a1-a6 models on the mixture of image and video data, as mentioned in Table 2.

## A.8 N-GRAM DISTRIBUTION

In this section, we compare the 2-gram and 3-gram distribution of dVAE (Ramesh et al., 2021), VQGAN (Esser et al., 2020) image tokeizers. We compute 2-gram and 3-gram distributions on the discrete tokens of 10000 ImageNet validation images. Figure 10 and Figure 11 show the distributions of these tokenizers respectively. On 2-gram distribution, dVAE (Ramesh et al., 2021) has more discrete combination of tokens compared to both VQGAN-1K and VQGAN-16k tokenizers.

## A.9 ATTENTION PROBING VARIANTS ON K400

We also evaluate our models and baselines on the Kinetics 400 dataset using a variant of attention probing. In the main paper, we use attention probing, with only learning $W_k, W_v$ matrices, and a single learnable query vector. We also test with cross attention with MLP layers as the attention classifier, to give more capacity to the learnable head. Table 17 show the performance on the attention classifier with an additional MLP head. This helps to performance improve across over all models.

| Model | Params | Dimension | Heads | Layers |
|-------|--------|-----------|-------|--------|
| a1 | 14.8M | 256 | 16 | 12 |
| a2 | 77.2M | 512 | 16 | 16 |
| a3 | 215M | 768 | 16 | 20 |
| a4 | 458M | 1024 | 16 | 24 |
| a5 | 1.2B | 1536 | 16 | 28 |
| a6 | 1.9B | 1792 | 16 | 32 |

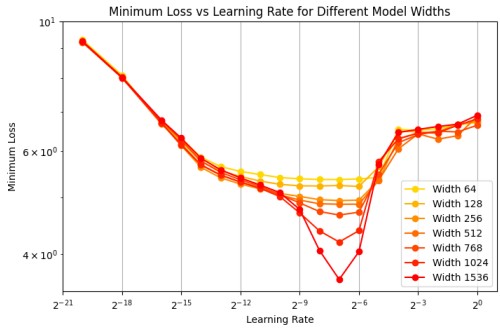

**Table 15: *Toto* Varients:** We scale *Toto* models by increasing hidden dimension and number of layers linearly while keeping number of heads constant following (Yang et al., 2022; Touvron et al., 2023).

**Table 16: $\mu$-Parameterization Learning Rate:** We show that $\mu$-Parameterization Yang et al. (2022), we can train all width *Toto* models, with an single optimal learning rate of $2^{-7}$.

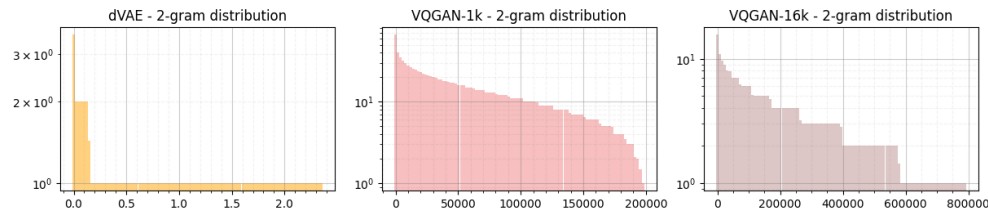

**Figure 10: 2-gram Distribution of Various Tokens:** We compute the 2-gram distribution on 10000 images from the ImageNet validation set. Compared to VQGAN 1k and 16k vocabulary tokenizers, the dVAE tokenizer has a larger set of token combinations.

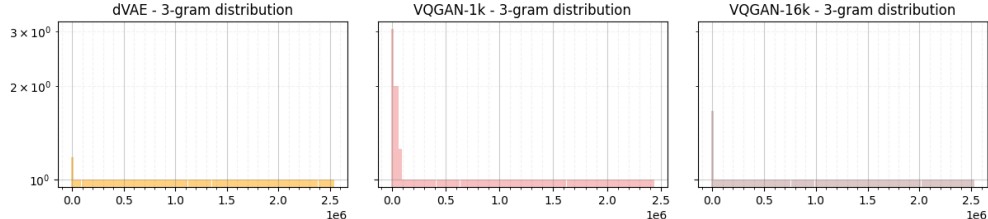

**Figure 11: 3-gram Distribution of Various Tokens:** We compute the 3-gram distribution on 10000 images from the ImageNet validation set. All the tokenizers has similar almost flat distribution when it comes to 3-gram tokens.

## A.10 VISUALIZING OBJECT PERMANENCE

We identify specific attention layers that attend to the last appearances of the currently generated object tokens (e.g. attention layer 15 in `1b` model). Visualizations of the attention coefficients of such layer for two videos with reappearing bottles are presented in 12. In the first video (left), the bottle with the sticker is occluded by a bag and reappears. In the second video (right), the bottle disappears and reappears due to camera motion. In both cases, when visualizing the attention coefficients for this layer for a token corresponding to the first reappearance of the stickers on the bottles, the only attended tokens are tokens from the same frame and tokens that correspond to the same sticker in the first frame. This suggests that the model learns temporal correspondences across the video.

| Method | Arch | Top1 |
|---|---|---|
| Hiera (Ryali et al., 2023) | Hiera-L/14 | 74.2 |
| Hiera (Ryali et al., 2023) | Hiera-H/14 | 75.2 |
| VideoMAE (Wang et al., 2023a) | ViT-B/14 | 65.4 |
| VideoMAE (Wang et al., 2023a) | ViT-L/14 | 74.8 |
| *Toto*-base | LLaMA | 61.2 |
| *Toto*-large | LLaMA | 65.8 |
| *Toto*-1b | LLaMA | 74.8 |

**Table 17: K400 Results:** We evaluate our models using cross attention and MLP layer as the classification head. Overall using a high-capacity head improves the performance across all models.

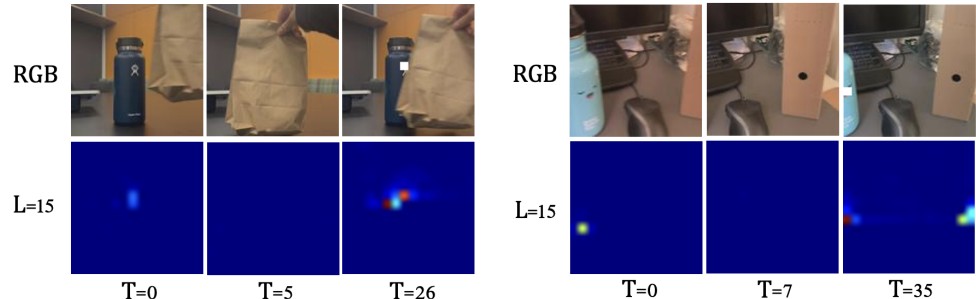

**Figure 12: Attention Visualization.** We manually identify layers in which the model attends to the previous occurrences of the token that the network generates (e.g. the tokens that correspond to the reappearing stickers on the bottles). Layer 15 sparsely attends to such tokens in the first frame in context, as well as nearby tokens in the current frame.

### A.11    GENERATION SAMPLES

**Solve tasks with generation: long video generation:** we can generate up to 64 frames, first raw: periodic motion, second raw: object permanence (light stand).

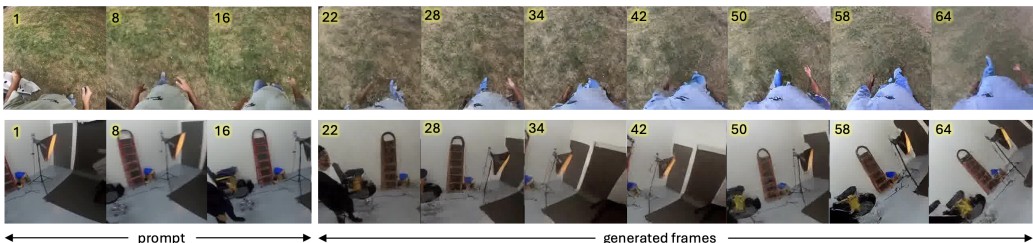

**prompting (pre-trained model):** shows 3D rotation

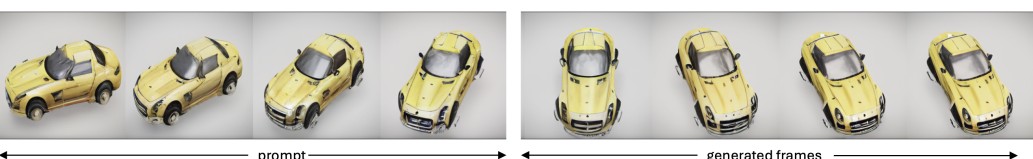

**prompting (finetuned model):** A small 1000-step fine-tuning leads to a promptable model for various vision tasks.

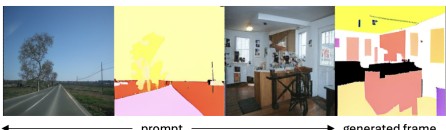 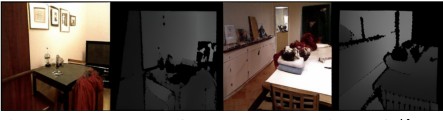

**More out-of-distribution samples:**

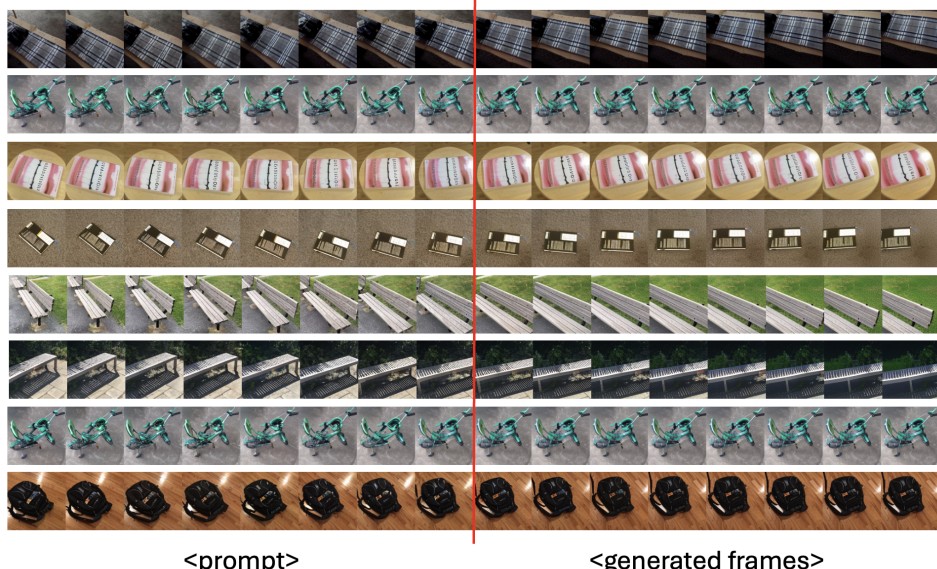

<prompt>                    <generated frames>

**Figure 13: Samples from CO3D dataset:** We prompt our model with first 8 frames from co3d dataset, and generate the rest 8 frames.

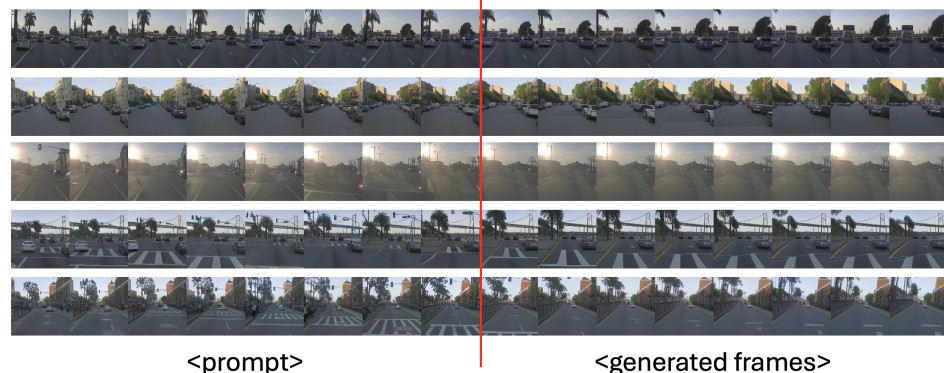

<prompt>                    <generated frames>

**Figure 14: Samples from pandaset dataset:** We prompt our model with first 8 frames from pandaset dataset, and generate the rest 8 frames.

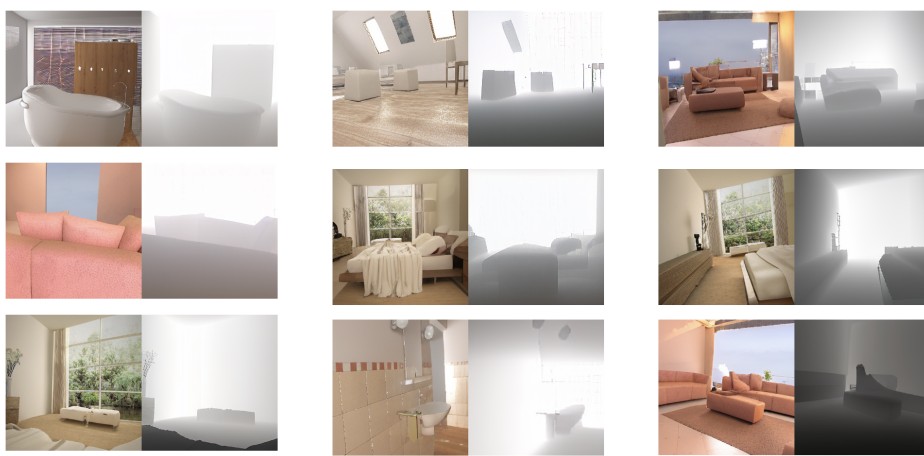

**Figure 15: Depth Estimation:** We finetune our model on depth estimation, given RGB frames as a prompt and generate the depth image.

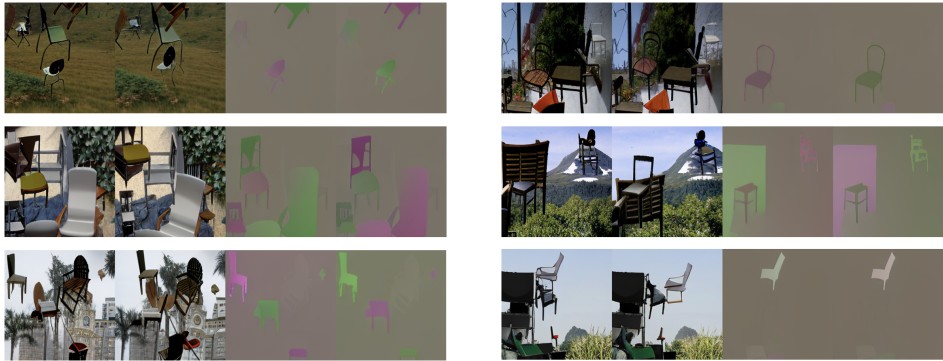

**Figure 16: Flow Estimation:** We finetune our model on flow estimation task, given first and second RGB frames as a prompt and we generate the flow image on x direction and y direction.

A.12   VISUALIZING FEATURES

We take our large model and use randomly selected 100 images from the ImageNet validation set to store the intermediate features from all layers. Then we compute PCA on each layer individually. After this, we use a few more images to get their intermediate features and compute their first 3 PCA components as in (Amir et al., 2021). Figure 17 shows the first three projections of the *Toto*-large model features.

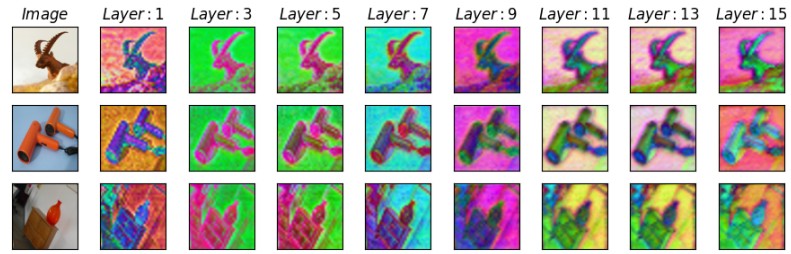

**Figure 17: Features Visualization.** First 3 PCA components of each layer of vGPT-large model features.

A.13   ADDITIONAL LAYER-WISE PROBING RESULTS

We also probe the multiple variants of our models at each layer for the best ImageNet performance. First, we test the models on linear probing, on both sizes of 128 and 256 resolution. We also plot the probing curves for the models trained with attention probing at 128 resolution. Across all these models, the performance has a similar behavior to the pre-trained models, with peak performance at the middle of the depth of the model.

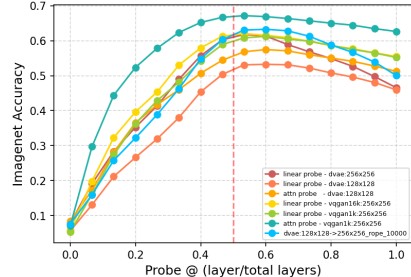

**Figure 18: Training Loss Curves:** We show the training loss curves for multiple variants of our models.