# OpenReview forum: "Token to Token Learning From Videos"
_ICLR.cc/2025/Conference — Submitted to ICLR 2025_

### Official Review · Reviewer_veSd · 2024-10-28

**Soundness:** 1
**Presentation:** 2
**Contribution:** 1
**Rating:** 1
**Confidence:** 4

**Summary:**

This is a tech report on failed attempts of using auto-regressive video token prediction objective for representation learning. In particular, the authors use the standard next patch prediction objective to train the LLaMa transformer model on the HowTo100m video dataset (with some data from other image and video datasets in the mix, but HowTo100m constitutes ~95% of the training data). They then evaluate the resulting representation on a variety of downstream task (image- and video-based, as well as robotics manipulation) with linear-probing and compare to a few baselines. Despite most evaluations being designed to favor the proposed approach (e.g. by comparing to smaller versions of baseline models or only reporting weak baselines) it still underperforms compared to the baselines in most experiments. Somehow, based on these results the authors clam that "despite minimal inductive biases, our approach achieves competitive performance across all benchmarks".

**Strengths:**

The paper is relatively well written and easy to follow for a tech report. For a research paper the overall presentation is not satisfactory (e.g. huge chunks of related work are omitted entirely).

A comprehensive ablation study of the proposed approach is reported.

**Weaknesses:**

There are major omissions in the related work overview (and, as a results, in the evaluated baselines). Most importantly, the paper completely ignores prior work on visual representation learning via image and video generation with diffusion. This is the closest research direction to the method proposed here, is widely adopted for downstream computer vision tasks (e.g. Namekata et al., ICLR'24, Zhu et al., ECCV'24), and thus would serve as the most natural baseline, but the authors never even mention it. In fact, for some of the tasks evaluated in this paper (e.g. video object segmentation) StableDiffsuion representation is the de-facto state of the art, outperforming the proposed approach by leaps and bound on DAVIS, but the authors conveniently do not compare to it.

Another critical omission is existing auto-regressive methods for video generation (which go at least as far back as Weissenborn et al., ICLR'20, more recently there's Language Model Beats Diffusion: Tokenizer is key to visual generation by Yu et al.). In fact, the authors go as far as to claim extending the auto-regressive objective to videos as their contribution. Consequently, the authors do not compare their auto-regressive video generation approach to the similar state-of-the-art methods neither in terms of video generation quality nor in terms of the downstream performance.

In the presented evaluations, when compared to the versions of the baseline objectives (e.g. MAE) with a comparable number of parameters, the proposed approach underperforms dramatically on most tasks (e.g. by ~10 accuracy points on ImageNet or by ~14 accuracy points on Kinetics). The only task on which a comparable performance to the state-of-the-art is demonstrated is action forecasting, which is hardly surprising given the proposed representation is trained with a forecasting objective. In fact, what is surprising is that even on this highly favorable task the propped representation doesn't show significant advantages over auto-encode objectives like MAE.

On some of the tasks the authors try to show competitive performance by only comparing to weak baselines. For example, on DAVIS they ignore sota approaches, and only compare to DINOv1 and MAE-base. DIFT (Tang et al., NeurIPS'23) outperforms the best version of the proposed approach by 13.3 points by capitalizing on the image diffusion objective (no video data in training). In the robotics experiments they only compare to the baseline representation from the paper that introduced the dataset in 2022, omitting even the representations evaluated in the other experiments in this paper (e.g. MAE or DINO). Same goes for the CATER experiments, only here the baseline is a ResNet from 2019.

As discussed above, the key claim of the paper that the proposed approach achieves competitive performance across all benchmarks is simply not true. There are also no technical contributions in this paper. In short, there is just no contribution.

**Questions:**

Please compare your approach to the state-of-the-art image and video generation representations (both via diffusion and via next token prediction) on all tasks.

Please compare the video generation quality of your approach to the state-of-the-art.

Please report actual sota representations on all tasks (e.g. by searching through the latest citations for the corresponding dataset papers if you are not aware of what these are).

What is the contribution of your paper?

---

> ### Author Response · Authors · 2024-11-17
> **Official Response by Author for the Reviewer veSd**
>
> We thank the reviewer for their feedback, and we have answered their questions from the reviews. We are happy to discuss this further and answer more questions to clear any concerns or do more experiments if this can help with the rebuttal.
>
>
> **Novelty**
>
> as stated in the abstract, we empirically study the generative pretraining on videos, this paper does not describe a novel method, instead, it studies a straightforward, yet must-know baseline given the recent progress of large language models pre-training for self-supervised vision pretraining. This paper does not describe a novel method, Instead, It studies a straightforward, yet must-know baseline given the recent progress of large language models pre-training for self-supervised vision pretraining. As the reviewer mentioned, the paper has the design choices and all the insights for next token pre-training for vision tasks and scaling laws. With the huge success from large language model pretraining, we believe this paper establishes a good baseline for video pretraining, and given all the design choices, we can build better models on top of this, and we will release the model weights and training code and we also careful with the data, and only used publicly available data sources for the pretraining for reproducibility purposes.  This has been done in vision before for example, in Chen et al (An Empirical Study of Training Self-Supervised Vision Transformers).
>
> **DAVIS experiments and comparison**
>
> For the DAVIS experiments, we respectfully disagree with the reviewer that DIFT is compatible with our approach, DIFT is a stable diffusion model, trained on image-text pair, and thus having learned binding via text supervision. On the other hand, we learn the model with no textual supervision. Having this textual supervision can help with binding problems and have better performance on tracking, but we should compare our approach with only self-supervised approaches.
>
> **Generative pre training objective**
>
> For the auto-regressive or generative objective, we only use this for pretraining, same as in large language models and iGPT (Chen et al). We only claimed we used generative objective, and we never claimed that we get better performance on video generation, and that was not the goal of this paper. In the same spirit as iGPT for example, we were only interested in learned representation quality using a generative objective.
>
> **Generative pre training objective**
>
> For the concern regarding the performance, our method is not state of the art on all the tasks, (eg, ImageNet and Kinetics on probing). For example on imagenet, linear probing with causal attention leads to drop in performance, however as we discussed in appendix A4, when finetuning our models on imagenet with full attention, we get similar results as all the vision encoders on same size. This has been observed in MAE also, that linear probing does not favor generative approaches compared to discriminative approaches.
>
> **Ego4D experiments**
>
> For the Ego4D tasks, our large model achieves best performance. As we look at all the numbers, we can see this task is hard and all the models struggle with this dataset. While our model still has gains on these tasks, and we did not claim this is the best performing model on this task, due to smaller gains, which we also agree with the reviewer.
>
> **Robotics experiments**
>
> Regarding the robotics experiments, we want to mention that MVP is a MAE model, and it has trained on egocentric videos and mvp paper showed that MAE trained on ego4d is better than MAE trained on imagenet.
>
> **Discussion on Weissenborn et al.**
>
> Thanks for the suggestion, and we agree this paper is related, and we will cite and discuss in the paper. We also want to note that this paper still deals with generative tasks on Kinetics and BAIR robot pushing. However, our work focuses on generative pretrained models for perceptual tasks.

---

> > ### Author Response · Authors · 2024-11-17
> > **Response -- continued**
> >
> > **“there is just no contribution”**
> >
> >
> > We strongly disagree with the reviewer’s statement. From Attneave (1954),  many steps are needed to achieve practical engineering success - the ChatGPT moment for video; iGPT was one of the steps, our work is another. We would like to repeat “This paper does not describe a novel method, Instead, It studies a straightforward, yet must-know baseline given the recent progress of large language models pre-training for self-supervised vision pretraining.`` Other reviews also agree with this and said Reviewer 5ayv: “The paper is extremely well written and presented. It was a pleasure to read”, Reviewer hFJf: “confident that this is a significant paper in terms of pushing discussion in the community forward re: generative vs discriminative pre-training for image and video models”. We analyze several design choices on tokenization, evaluate on a wide range of experiments, and we will release the models, training and eval codes for further research. We believe these findings are extremely valuable for further research on scaling vision models. We also quantify the scaling laws for our models and show that they don't scale as language models. This serves as a strong baseline to build pure vision models which can scale. More specifically, we showed even with discrete tokens we can pre train at low-resolution and fine-tune at high resolution with RoPE embeddings to get better performance (Table 3), with decoder only model, we showed that best performance is happening in the middle of the model depth for all tasks and all model sizes (Fig 9), For fine tuning, we showed that if the model is pre trained with videos, it can still be finetuned with full attention without loss in performance (table 13), We also showed the benefits mu-parameterization for visual next token prediction tasks (Table 16).
> >
> > **Comparison with diffusion models**
> >
> > We kindly disagree with the request for comparing our work to other generative approaches. Our work is a generative pretrained approach for self-supervised learning, while other DIFT for example utilize image-text papers, and video-poet for example again trained with video-text pair. It is not reasonable to compare a fully self-supervised approach to a model trained with paired data (which is close to learning from noisy labeled data and can learn binding easily).
> >
> > For the video generation quality, the goal of this paper is not video generation, we used generative pretraining to learn a good representation, not for producing good videos.

---

> > > ### Comment · Reviewer_veSd · 2024-11-20
> > >
> > > I thank the authors for their detailed response. In this response the authors have confirmed the main criticism of my review. Specifically, the only contribution of the paper is in providing a public implementation of a well known baseline and ablating some of its hyper-parameters. This perfectly describes the scope of a tech report, not of a research paper. The authors are encouraged to convert this submission into a blog post, which might indeed have some value for the community.
> > >
> > > Regarding fairness of comparison to DIFT and StableDiffusion more broadly. There are two valid ways to write a paper like this. The first one involves treating all existing representations as "foundational models" and proposing a new one that is superior to them. In this scenario such factors as the exact nature of the training data used by each method or even their model sizes can be disregarded to an extent. In the second approach a new self-supervised learning objective is proposed and compared to existing objectives in a controlled environment. This, of course, requires using the same (or at least very similar) model sizes and data sources for all the studied variants on all tasks. In this work the authors are taking neither of these paths, and are instead disregarding differences between models when it's convenient to them (e.g. when comparing to much smaller variants of the baselines in the paper, or comparing to baselines trained on a lot less data), and are using these differences to avoid comparing to strong baselines in other scenarios (e.g. by refusing to report DIFT because it is based on an visual-language representation). Needles to say, this is not acceptable.
> > >
> > > Finally, diffusion objective has nothing to do with language fundamentally, and image2video diffusion models can be trained in the same setup as what's proposed in this work. The authors have to compare to video diffusion (which is a superior approach to video generation) if they are advocating using next token video prediction for representation learning.

---

> ### Comment · Reviewer_5ayv · 2024-11-20
>
> This review is overly harsh and I would strongly suggest that the reviewer take a less personal tone. Beginning the review with "This is a tech report on failed attempts of using auto-regressive video token prediction objective for representation learning" is almost crossing the line for me in terms of basic etiquette. One can make constructive criticisms about a paper *without* being mean and assuming deceptive intent from the authors.
>
> However, I do think that there are some valid concerns about the experiments in the paper. This paper does not claim to invent autoregressive video generation for representation learning - my understanding is that it attempts to carefully validate and measure the effects on several different tasks. This work is valuable - too much of today's computer vision community abandons careful analysis that can help many researchers for "novelty" that will never be used.
>
> Given that, the experiments in this paper must be thorough and ironclad and the resulting analysis should provide useful insights for the community. I'm not entirely sure this is the case yet, and I would encourage Reviewer veSD and the authors to continue to discuss this more agreeably.
>
> I do agree with Reviewer veSD concern's about missing works, but most of these works are really not comparable. This paper is about self-supervised pre-training, while many of the works listed as missing are not self-supervised! This would be an unfair and not particularly helpful comparison. This is also partly on the authors for not making it clear why these baselines are not included - I think it is important to explicitly note this in the text.
>
> I think comparing to diffusion (which is trained on paired video-text data) is also somewhat nonsensical for the paper here. The point is to do next-token prediction only and measure the impact from this! I'd really appreciate Reviewer veSD's comments on why they believe this is absolutely necessary.

---

> > ### Comment · Reviewer_veSd · 2024-11-21
> >
> > Dear reviewer, thank you for your opinion.
> >
> > The first sentence of this review is meant as an "executive summary" for the AC, hence the terse form. The statement itself is accurate and is substantiated with references and quantitative comparisons in the rest of the review.
> >
> > Regarding the necessity of comparison with the video diffusion objective. This submission advocates using video generation for visual representation learning. Diffusion, not next token prediction, is the dominant approach for video generation in the literature at the moment. Hence comparing with video diffusion is necessary to judge the effectiveness of the proposed approach for visual representation learning via video generation. Another critical baseline/related work that is omitted is V-JEPA [1], which also uses future token prediction in videos for representation learning and achieves superior results compared to the variant of this objective proposed here.
> >
> > Moreover, text-conditioned image/video diffusion is, arguably, the most effective visual representation learning paradigm at the moment. Rather than ignoring it, the authors should add text conditioning to their model (which is trivial to do given it's based on an LLM), train it together with language annotations avaialbe in HowTo100m, and compare with the actual stat-of-the-art visual representation learning approaches, not with outdated ones like VAE or DINO.
> >
> > [1] V-JEPA: Latent video prediction for visual representation learning,
> >   Bardes, Adrien and Garrido, Quentin and Ponce, Jean and Chen, Xinlei and Rabbat, Michael and LeCun, Yann and Assran, Mido and Ballas, Nicolas

---

> ### Author Response · Authors · 2024-11-22
>
> We appreciate the feedback from reviewer **5ayv** regarding the irrelevance of vision-language pretraining and diffusion objectives to our work. Below, we address the specific concerns raised by reviewer **veSd**:
>
> ### Contribution of the Paper
> **Reviewer Comment**: *"The only contribution of the paper is in providing a public implementation of a well-known baseline and ablating some of its hyper-parameters."*
>
> **Response**: We strongly disagree with this statement. Our paper goes beyond merely ablating hyper-parameters. It provides a comprehensive study of a crucial baseline in the context of recent advancements in large language model pre-training for self-supervised vision pretraining. Other reviewers, such as **5ayv** and **hFJf**, have recognized the value of our work. Reviewer **5ayv** noted the extensive evaluation across diverse tasks, and **hFJf** highlighted the significance of our paper in advancing community discussions on generative vs. discriminative pre-training.
>
> We analyze several design choices, evaluate a wide range of experiments, and will release models, training, and evaluation codes for further research. Our findings are crucial for scaling vision models, and we quantify scaling laws, showing that they differ from language models. Specific contributions include:
> - Demonstrating pre-training at low resolution with discrete tokens and fine-tuning at high resolution using RoPE embeddings for improved performance (Table 3).
> - Showing optimal performance in the middle of model depth for all tasks and sizes with a decoder-only model (Fig 9).
> - Proving that models pre-trained with videos can be fine-tuned with full attention without performance loss (Table 13).
> - Highlighting the benefits of mu-parameterization for visual next token prediction tasks (Table 16).
>
> These experiments, being novel, provide valuable empirical results, even if some are negative.
>
>
> ### Regarding DIFT
> As reviewer **5ayv** also mentioned, comparing a self-supervised approach to a supervised one (based on language) is unreasonable. We will include this work in the related works section.
>
> ### Diffusion Objective
> We kindly ask the reviewer to read our rebuttal. We never stated that diffusion objectives are related to language. Our work focuses on generatively pre-trained self-supervised learning, unlike DIFT or VideoPoet, which use image-text or video-text supervision. Comparing our fully self-supervised approach to models trained with paired data is not reasonable.
>
> ### Related Works
> We will address related works on diffusion-based and contrastive approaches in the revised paper. As **reviewer 5ayv** mentioned, our paper is about self-supervised pre-training, and comparing it to non-self-supervised methods would be unfair and unhelpful. We will clarify this in the related works section.
>
> ### Video Generation for Visual Representation Learning
> This submission focuses on a generative **objective** for **self-supervised** visual representation learning. There are no diffusion models achieving comparable results on ImageNet or similar benchmarks. If reviewer **veSd** is aware of such works, please share them. The closest we found is Hudson et al. [1], which achieves 72.1% on ImageNet, below our results. We will add this work to the paper.
>
> [1] SODA: Bottleneck Diffusion Models for Representation Learning
>
> Regarding V-JEPA, we have included I-JEPA and are willing to add V-JEPA along with other contrastive approaches. Reviewer **veSd’s** statement that "V-JEPA [1] uses future token prediction in videos" is incorrect. It is a contrastively trained method, not generative, and does not predict pixels. We have extensively discussed the differences between contrastive and generative approaches in the paper.
>
> ### Text-Conditioned Image/Video Diffusion
> We respectfully disagree with the suggestion to focus on text-conditioned visual representation learning. Scientific discovery requires exploring new directions, not just following the strongest current approach. Our paper's objective is to study **self-supervised** pre-training with **next token prediction** models, which we have done extensively. Training a text-conditioned next token prediction model is beyond our scope. We ensured no supervised signals were used throughout our work, from data curation to tokenization and pre-training.
>
> ---
>
> We hope this response clarifies our contributions and addresses the concerns raised. Thank you for your consideration.

---

> > ### Comment · Reviewer_veSd · 2024-11-22
> >
> > I kindly ask the reviewers to read my responses. When talking about diffusion as a necessary baseline, I was not talking about any specific method (e.g. DIFT), but rather about the diffusion objective itself. If you are advocating using image-conditioned video generation for visual representation learning then you should compare your next token prediction objective to the dominant approach to image-conditioned (language-free) video generation, which is diffusion.
> >
> > SODA is an image-based diffusion model, so it's unfair to compare it to the video-based model proposed here. There are many image2video diffusion models in the literature (e.g. RIN [Jabri et al., ICML'23], but there are certainly more recent and more effective ones). For a proper comparison, the authors should just take the most recent model with a public implementation, instantiate a variant of comparable model size, train it on HowTo100m and compare to their token prediction variant.
> >
> > Please see my discussion above about the evaluations in this paper being overall inconsistent. If the authors are suggesting that their work is a principled study of different self-supervised objectives, then all the objectives need to be compared under the same conditions (same model size, same type and scale of training data etc). This is not what's happening. If they argue that differences like this can be disregarded, then they should compare to text-conditioned models as well.
> >
> > Text-conditioned generation is not equivalent to supervised learning. I do agree that it is not fully comparable to image-conditioned video generation. Same as image-based methods like DINO or SODA are not fully comparable to the video-based approach proposed here, which does not stop the authors from including them as baselines.
> >
> > V-JEPA is not contrastively trained. Instead an L1 loss is used to predict latent tokens in videos. This is not the same as the next token prediction objective proposed here, but is highly relevant and 100% comparable (modulo model size, which is generally smaller for V-JEPA). To re-iterate, that approach strongly outperforms the one proposed here.

---

> > > ### Author Response · Authors · 2024-12-04
> > >
> > > **“If you are advocating using image-conditioned video generation for visual representation learning then you should compare your next token prediction objective to the dominant approach to image-conditioned (language-free) video generation, which is diffusion.”**
> > >
> > > **Thanks to reviewer HFJF**, we added more related works regarding generative models for representation learning.
> > >
> > > *Generative Objectives: Apart from the next token prediction objective, Diffusion generative pre-training also learns meaning full representations[1]. On classification tasks, Ayromlou et al, and Shipard et al showed that using diffusion models for creating data augmentations and synthetic data can help with classification tasks. Mukhopadhyay et al (CDG), achieve 71.9% on imagenet 1k with fully self-supervised diffusion objective without text condition. Mukhopadhyay et al (DifFormer) achieve 76.0% in imagenet 1k, by learning an attention head to fuse the features from the diffusion model. Xiang et al (DDAE) using an unconditional diffusion model’s features yields competitive results on CIFAR10 and TinyImageNet. SODA (Drew et al), DAE (Chen et al) and DiffMAE (Chen et al) all show reasonable performance on Imagenet, learning an encoder and using a diffusion based decoder model. On the other hand DiffusionClassifier (Li et al) prompts a pre-trained diffusion model to find few shot classification by measuring the reconstruction loss. JDM (Deja et al), jointly learns a diffusion model and representations for classification. HybViT (Yang et al) shows vit can be used as generative models and shows both generative and representation features of the hybrid model.*
> > >
> > >
> > > **"For a proper comparison, the authors should just take the most recent model with a public implementation, instantiate a variant of comparable model size, train it on HowTo100m and compare it to their token prediction variant."**
> > > — This is beyond the scope of this work, there have been research of how to get recognition tasks solved with diffusion models, SODA was one, Chen et al (Deconstructing Denoising Diffusion Models for Self-Supervised Learning) is another attempt, and reached only 70.9%. Solving recognition tasks with Diffusion models, itself is an open research question and this is not the scope of this paper. However, we have added a clear discussion on related works with diffusion models and we will include this in the paper.
> > >
> > > **“If the authors are suggesting that their work is a principled study of different self-supervised objectives, then all the objectives need to be compared under the same conditions”**
> > > — We kindly ask the reviewer to read the paper and rebuttals, and please do not make statements on your own. We never said that our work is a “study of different self-supervised objectives”.
> > >
> > > **“V-JEPA is not contrastively trained.”**  —- We kindly ask the reviewer to read [1], “Without involving negative instances, BYOL trains the online network from an augmented view of an image to predict the target network representation of the same image under a different augmented view (positive instance).” as stated in [1]. As shown in Figure 1 of [1], Both JEPA and Contrastive methods operate similarly, even if the models do not have negatives.
> > >
> > > [1] Shentong Mo, Shengbang Tong. Connecting Joint-Embedding Predictive Architecture with Contrastive Self-supervised Learning. Neurips 2024

---

### Official Review · Reviewer_cPsN · 2024-10-29

**Soundness:** 3
**Presentation:** 1
**Contribution:** 2
**Rating:** 3
**Confidence:** 3

**Summary:**

This paper presents an approach for generative pre-training from videos, and collect a large video dataset and conduct a large-scale empirical study across a range of diverse tasks.

**Strengths:**

1. This paper makes architectural improvements on generative pre-training to enable scaling to videos.
2. The approach achieves competitive performance across all tasks.

**Weaknesses:**

1.The logic of the abstract is confusing, the author only explains what he has done, and the motivation and purpose are not clear enough. The introduction part is too short.
2.The connotations of the evaluation indicators in Tables 7 and 10 need to be explained. Some of the test indicators in the table are illustrated, e.g. J, F in Table 10.
3. Tables 7 and 8 show that the Top1 indicators are not convincing in performance, for example, they are not even as high as the DINO using ViT-B. Please explain that their performance is still meaningful, although they do not exceed some of the existing methods.

**Questions:**

1. Whether the details of the datasets collected in the article can be explained in detail, it seems that it is just a mixture of several datasets. Please explain in detail how the selection criteria for the dataset, the pretreatment step, or the mixture were determined.
2. Please elaborate on the problem definition, application background, hardware specifications used for the experiment, and other details.

---

> ### Author Response · Authors · 2024-11-17
> **Official Response by Author for the Reviewer cPsN**
>
> We thank the reviewer for comments on the detailed experiments. Here we answer the questions from the review, and we are happy to answer more questions or get more experimental results to clean any concerns.
>
> **“Motivation and problem statement in abstract”**
>
> We will modify the abstract and explicitly state our motivations and contributions, along the lines of, “This paper does not describe a novel method, Instead, It studies a straightforward, yet must-know baseline given the recent progress of large language models pre-training for self-supervised vision pretraining.”
>
> **Questions regarding performance**
>
> Regarding the performance in Table 7, DINO-base model, we have grouped them into generative and discriminative approaches in Table 7, it has been shown that for probing discriminative models dominate in terms of performance. However, if we do full finetuning, as in appendix A.4 table 13, we can see that on our base model is achieving similar performance as DINO when fully fine tuned.
>
> **Data mixture**
>
> For the mixture of the datasets, one of the main criteria for the data is to use publicly available video datasets, for reproducibility. Apart from that, we also wanted to make sure the model should learn from egocentric and exocentric videos. Addition of images was to ensure the data diversity compared to video frames. We agree that we did not ablate the optimal ratios for the mixture of mini batch sampling. However, we would like to explore more systematic approaches for optimal ratios.
>
> **Evaluation Metrics**
>
> We elaborate about the problems and hardware:
>
> Semi supervised tracking: For the evaluation we used the standard metics Jaccard index J, contour accuracy F, and their average J &F (A benchmark dataset and evaluation methodology for video object segmentation, Perazzi et al)
>
> Robot manipulation: We reported the success rate, on both simulation and real world experiments.
>
> Video forecasting: This task requires identifying the bounding box of objects to be interacted with. Per bounding box, we assign a class label to the object (noun) and type of interaction (verb) as well as the time to contact (ttc) between the last seen frame and the future hand-object contact frame. We report the Top-5 Mean Average Precision of this task.
>
> Image recognition: We report top-1 accuracy.
>
> Video recognition: We report top-1 accuracy with the protocol in slowfast, where we sample multiple temporal segments and get prediction scores.
>
> **"Please elaborate on the problem definition, application background, hardware specifications"**
>
> We study the next token prediction task as self-supervised pre training for  vision models. We pretrain our models on 256 TPUs and models are evaluated on a wide range of tasks. Finally, we also established a large scale experiment, to quantify scaling law for video next token prediction models.

---

### Official Review · Reviewer_hFJf · 2024-11-05

**Soundness:** 3
**Presentation:** 3
**Contribution:** 4
**Rating:** 8
**Confidence:** 4

**Summary:**

This paper proposes a causal autoregressive model that jointly learns on discrete image and video tokens (videos tokenized independently per frame). When trained on 1 trillion tokens sampled from ImageNet, Kinetics-600, Ego4D and HowTo100m, the model shows competitive downstream performance on ImageNet classification, action recognition on Kinetics, action anticipation on Ego4D and Video Segmentation Tracking on DAVIS-2017 compared to relevant baselines. The authors also show the emergence of a scaling law when training their joint model on images and videos.

**Strengths:**

- Originality

The proposed method is original in training a pure decoder only causal AR model on images and videos without any conditioning. The authors however do claim using discrete tokens and RoPE embeddings as contributions which can be considered not exactly original. Their use in this exact setting, however, is novel.

- Quality

Experiments are clearly defined and seem to be well executed. Not all decisions are thoroughly studied, such as the data used and data mixture per minibatch, or the efficacy of VQGAN tokenizer beyond Imagenet-1k results. The authors claim to study tokenizers in detail, however the study only scratches the surface (discussed later). However, given the computation budget required for these experiments, I believe these could be considered as good-to-have but not must-have.

- Clarity

The paper is written very clearly, barring minor writing errors and missing information.

- Significance

I am confident that this is a significant paper in terms of pushing discussion in the community forward re: generative vs discriminative pre-training for image and video models. They show diverse results (though mostly in-domain w.r.t. training data for quantitative results) and do not over-claim performance.

**Weaknesses:**

- Tokenizer study

The authors only study image tokenization. No video tokenization is attempted or discussed. Within image tokenization, the authors study discrete tokenizers using dVAE or VQGAN. They conclude that various tokenizers have little effect on ImageNet top-1 readout accuracy. However, there is more to unpack here. VQGAN can achieve the same performance using 16x16 tokens and 1k vocabulary as dVAE does with 32x32 tokens. This is not discussed. dVAE with 16x16 tokens and 8k vocabulary performs ~8% worse than VQGAN with 16x16 tokens and 1k vocabulary. The authors choose dVAE justify choosing dVAE due to its tokens showing higher coverage (usage of 1-gram, 2-gram tokens across data) over VQGAN-16k, which is reasonable, but definitely not enough to justify using dVAE given the ImageNet readout result. Showing results for ToTo trained with VQGAN tokenization beyond Imagenet-1k would be very appreciated and would shed more light into this design choice. VQGAN tokenizers at 32x32 tokens are not tested at all, or discussed.

- Related Work

The related work section is sparse, especially in the Robotics section. Video Prediction has been studied in Robotics as pre-training and needs to be discussed. For exampleL Wu et al. ICLR 2024: Unleashing Large-Scale Video Generative Pre-training for Visual Robot Manipulation.
AIM, El-Nouby et al. ICML 2024, is a very related work that is only mentioned twice in the paper. No comparison results are shown despite AIM code and weights being available. The only discussion for AIM is that it trains on images only, using patch embeddings using data filtered using data filtering networks. In the abstract and conclusion, the authors say they use discrete tokens and RoPE embeddings as compared to prior work, which I must presume to be AIM since the authors do not cite any prior work with those sentences. The lack of comparison and mention of AIM across the paper needs to be improved.
Vision-Mamba is not discussed at all.
There is almost no discussion of diffusion based models and their use as generative pre-training methods. This is also an oversight in my opinion, since they are being used as vision backbones for perception, notably Marigold-depth (CVPR 2024).

- Comparison with Mamba

The comparison to Mamba is unclear. Which model was used exactly? The table cites the original Mamba paper which does not have vision experiments as far as I know. Did you train your own model? What were the design choices used? How is it related to VisionMamba (Zhu et al. https://arxiv.org/abs/2401.09417)?

- Major experiments are in-domain

The major experiments in the paper are in-domain. The model was trained on ImageNet, Kinetics and Ego4D and later tested on those three datasets. The DAVIS experiment is out-of-domain and novel and appreciated. The robotics examples are simple and only compare to Masked Visual Pre-training (Radosavovic et al.). There are interesting experiments in the supplementary material on adapting the method for semantic segmentation, depth prediction and video generation. It would have made the paper much stronger to have evaluated those tasks as a sign of the strength of joint vision encoding and generation, which parallels recent efforts of repurposing diffusion based image and video generative models for perception.

- Ablating data

How was the data mixture per minibatch decided? How was the dataset decided? On that note, the authors in the conclusion mention they "collect" a large video dataset. This is misleading since they mix existing datasets together.

**Questions:**

Please address concerns from the weakness section.

- Unclear details in Action Forecasting results
How were tokens at 5 layers extracted and fused using the pyramid network from StillFast? Please add this detail to the paper or supplementary and in the rebuttal.

- Figures and writing
The writing refers to GPT-2 as GTP-2 at times. Figure 8 caption is copy pasted from Figure 4 and not updated. Section 4.8 about scaling laws refers to Figure 16 from supplementary, whereas it should refer to Table 16 from Supplementary.

I am rating this paper as a 8, but really I see it as a 7 due to the concerns raise in the weaknesses section. I applaud the authors for their great work.

---

> ### Author Response · Authors · 2024-11-17
> **Official Response by Author for the Reviewer hFJf**
>
> We thank the reviewer for the valuable comments. Especially we thank the reviewer for their comments “well executed experiments”, “paper is written very clearly” and “confident that this is a significant paper in terms of pushing discussion in the community forward re: generative vs discriminative pre-training for image and video models”. We really appreciate your comments and here we answer your questions, and we are happy to answer any more questions or run experiments if it would be helpful with the rebuttal.
>
> **Question regarding novelty/originality**
>
> For the novelty, yes we agree with the reviewer. As stated in the abstract, we empirically study the generative pretraining on videos, this paper does not describe a novel method, instead, it studies a straightforward, yet must-know baseline given the recent progress of large language models pre-training for self-supervised vision pretraining. This paper does not describe a novel method, Instead, It studies a straightforward, yet must-know baseline given the recent progress in computer vision and large language models pret-training for self-supervised vision pretraining. As the reviewer mentioned, the paper has the design choices and all the insights for next token pre-training for vision tasks and scaling laws. Our main focus was to establish the visual only next token prediction as a scalable self-supervised video pre-training for vision models, with minimal inductive bias and no supervision at any stages: from data curation to preprocessing to pre-training.
>
> **“Discussion on data mixing ratios”**
>
> For the data mixture, we agree with the reviewer, we did not ablate the mixture of the optimal ratios for best performance, mainly due to compute limitations. Our mini batch was sampled with 40% howto100m, 20% ego4d and 20% kinetics, and 20% imagenet images. However , we are happy to add more small scale ablations on data mixtures, if the reviewer thinks this would help with the rebuttal.
>
> **“Reasons for choosing dvae over vqgan”**
>
> For the tokenizer study, apart from the results in table 3, a main other reason to choose dvae over VQGAN was the training of vqgan involves perceptual loss for alex net or vgg. This implicitly contains category information, and we wanted to build this model with zero supervision. We agree this might have hurt on some performance metrics, and we embrace it. This builds a clear self-supervised baseline for next token video models and further research can build on top of this and build better models with extra supervision or inductive bias. We are happy to run any specific experiments to give more insights (on a relatively smaller scale) during the discussion period.
>
> **“Related works on robotics”**
>
> For the related works section on robotics, we will cite and discuss Wu et al, it is very related to toto. Both works utilize the large scale generative video pre training for learning robot manipulation skills. One main difference between wu et al and ours is, for our pre training we only utilize vision data, while wu et al learn from vision-language data. While we should not discriminate data, in this work we took a stand to build toto only with vision modality.
>
> **“Discussion on AIM”**
>
> We agree that AIM is related work. We have added AIM in the comparison table 7, and we will discuss more about AIM in related works. Ideologically both works move towards learning from next token prediction as a pretraining, with three main differences - a) ours more focused on video, b) we trained with cross entropy loss vs MSE loss for AIM c) having discrete tokens allows us to generate videos to study properties like object permanence. Use of prefix attention was a benefit from AIM when trained only on images, however we found that with video pretraining we can train with causal attention and at fine tuning we can use full attention without loss of performance. We were regarding iGPT (Chen et al) as “use discrete tokens and RoPE embeddings as compared to prior work”, it was the first line of work that showed promising results for visual representation from generative pretraining.
>
> **“Discussion on vision-mamba”**
>
> Thanks for the suggestion, we have added this and mamba in the related works, and a few differences to note are a) toto is self-supervised model, pre-trained with next token loss, and visionMamba is a supervised model trained for imagenet and coco detection and b) we focus is more on video data while visionMamba focuses on image only tasks.
>
> **“Diffusion based vision backbones”**
>
> There has been recent works on diffusion based approaches for perception such as Marigold to repurpose large scale vision pretrained models for perceptual tasks. Both diffusion and AR models have the benefit of iterative computation at test-time. While diffusion models have shown good success on pix-to-pix tasks, getting good representation from them is still not fully explored.

---

> > ### Author Response · Authors · 2024-11-17
> > **Response -- continued**
> >
> > **“Mamba experiments”**
> >
> > Indeed, we trained our own model mamba and matched the model parameters to the toto-base model, with our tokenized images. For how vision mamba related, vision mamba is a fully supervised approach, with bidirectional attention to train a vit style model for classification and other vision tasks. While the mamba model we used, was for pre training with causal masking. This way all the models in table 6 are pre trained for the same number of epochs for next token prediction tasks, and linear probed for image net classification.
> >
> > **“In domain experiments”**
> >
> > As the reviewer mentioned, we have the generation based qualitative results in the appendix. We note that our work focuses on visual generative objective pretrained models for solving visual perception problems. We will quantitatively evaluate a few of these datasets for visual perception tasks and report before the end of the discussion period.
> >
> > **“Data mixtures and curation”**
> >
> > Thanks for pointing out the data “collection” wording, we will remove that line from the paper. We only curated publicly available datasets, also for the main reason for a reproducible baseline. We don't have ablations on the optimal ratio of data mixtures for optimal model performance. partly because of the expensive compute it requires to do the search, for example we used mu-parameterization, so that we don't need to do hyperparameter search with larger models. We explore the optimal data mixture ratios.
> >
> > **“Action forecasting explanation”**
> >
> > About Action Forecasting results, Ego4D short-term action anticipation v1 task requires models to predict future actions from past context. We use our models as the backbone for the pyramid network used in StillFast. In accordance with their protocol, we apply linear probes to tokens at 5 evenly spaced layers of toto, analogous to different layers of their X3D backbone, and pass these new features through the Pyramid network. We fully fine-tuned our model with self-supervised next-patch loss along with task-related losses, and we observed having self-supervision loss improves overall performance. Table 9  shows the performance of our toto-large model on the Ego4D short-term action anticipation task. This task requires predicting the object to be interacted with (noun) and the type of interaction (verb) as well as time to contact (ttc) from the last seen frame to an estimated time between object-hand contact.
> >
> > Thanks for pointing out the typos and errors in the captions, we have updated the paper with these modifications.

---

> ### Comment · Reviewer_hFJf · 2024-11-26
>
> Thank you authors for the detailed response.
>
> As alluded to by the other reviewers, this work is not comprehensive in its comparison to related work and is not a standalone resource to further the debate on generative vs. discriminative pre-training, nor for next-token prediction vs. diffusion objectives. Of course studying these more deeply would have improved this paper dramatically. However, I disagree with the other reviewers that this is necessary for this paper to be accepted since the authors have been careful with their claims and are indeed proposing a baseline that while according to the other reviewers might be well known within their circles, it is not known in published literature.
>
> Re: the rebuttal, I would urge the authors to make the Mamba experiment clearer in the paper to enable clear future comparison. The revised version is still lacking enough detail for one to reproduce the experiment.
>
> Re: in-domain experiments, I also do not see new quantitative results beyond the qualitative results provided in the supplementary material. Please work on adding these. The current experiment suite being mostly in the domain of the pre-training data limits the impact of this paper for me.
>
> The discussion in the related work is still quite sparse and barely does justice to the papers mentioned. For example, saying that "however, AIM trains on data-filtering networks with clip filtered data" doesn't enough context.
>
> The choice of choosing DVAE over VQGAN is still not strongly reasoned. Arguing that the VQGAN pollutes the model with imagenet labels and might lead to lower performance on imagenet classification only goes so far since the quantitative experimental data is in-domain to the training data anyway. However, the tradeoffs between the choice of tokenizers were explored, which are useful to the community.
>
> The authors say in the rebuttal that "While diffusion models have shown good success on pix-to-pix tasks, getting good representation from them is still not fully explored". This is a factually wrong statement. Please start by looking at this survey paper: https://arxiv.org/abs/2407.00783 and add a discussion to the paper. This has been mentioned by the other reviewers as well, and even though I think the experiments with diffusion are not a "must-have", a discussion is a "must".
>
> I acknowledge that the answers to the rest of the questions are satisfactory and given the strong negative reaction from Reviewer veSD, I will be keeping my score at 8. However, I would like to reiterate that I really see this paper as something between 6 and 8.

---

> > ### Author Response · Authors · 2024-12-04
> >
> > **in-domain experiments**: We also added new quantitative results in the appendix. Generated video samples from driving data, and co3d data. Both of these are out-of-domain samples. In addition to this we also finetuned the model on other modality tasks such as depth estimation and optical flow estimation. In depth estimation where the first frame is RGB and the next frame is the depth map. For the flow estimation, the first two frames are from time step t and t+1, and the next to generated samples are for the flow in x direction and y direction. All the qualitative samples are in Fig 13, 14 , 15 and 16.
> >
> > **Results on positional embeddings**: We apologize for the delay in getting these results due to computer limitations. We have trained a model on only absolute positional embeddings. As in Table 4, we first train a LLaMA model with absolute positional embeddings, and it performs almost the same as LLaMA with RoPE. However, when fine tuning this model for larger resolution, we see that absolute positional embeddings do not match the performance of RoPE based context length extension.
> >
> > | Model   | RMSE   |
> > |---------|--------|
> > | LLaMA-RoPE-dVAE/16  | 53.2 |
> > | LLaMA-RoPE-dVAE/16->32  | 64.4 |
> > | LLaMA-Abs-dVAE/16  | 53.4 |
> > | LLaMA-Abs-dVAE/16->32  | 62.7 |
> >
> > **Mamba experiments**: We used a hidden size of 1536 and the 16 hidden layers and a vocabulary size of 8k. We trained this model for 400 epochs with batch size of 2 and gradient accumulation of 2. The model is trained with a peak learning rate of 1.25e-4 and weight decay of 5e-2. We also used a warm up of 2000 steps. We will release the code and models for full reproducibility.
> >
> > **Results on VQGAN**: We also conducted relatively smaller scale experiments on VQGAN as a tokenizer. First we trained two VQGAN models with and without perceptual loss. These models have an rfid score of 2.9 and 35.8. Then we use these as the tokenizer for training our toto-small model (hidden_size=384, layers=8) for 50 epochs. Model using perceptual loss tokenizer achieved 37.8% on imagenet classification and the model using the tokenizer without perceptual loss got 28.3%. We think without perceptual loss it might need longer training for both tokenizer and the next token prediction model to converge. We will add longer training and more ablation along this lines to study different losses involved in the training.
> >
> > **Related works**: We have updated the paper and revised the related works as recommended by the reviewer. We will also add a more in-depth discussion on generative pretraining.
> >
> > *Autoregressive Modeling: For Autoregressive pre-training, PixelCNN and PixelRNN} proposed generating pixels one by one using convolution  and bidirectional LSTMs. With the introduction of the transformers, ImageTransformers showed generating pixels with causal local attention performs better than previous CNN and RNN-based methods. While all of these methods focused on the generation quality of the pixels, iGPT~\citep{chen2020generative} showed that generative pre-training is also a good way to learn strong visual representations for recognition tasks. Recently D-iGPT revisited the ideas behind iGPT by using clip tokens. AIM on the other hand uses patch embedding rather than any pre-trained models for tokenization, however, it trains on Data Filtering Networks with clip filtered data. VisionMamba also showed how to utilize sequence models with bidirectional state-space modeling for supervised vision tasks.*
> >
> > *Generative Objectives: Apart from the next token prediction objective, Diffusion generative pre-training also learns meaning full representations[1]. On classification tasks, Ayromlou et al, and Shipard et al showed that using diffusion models for creating data augmentations and synthetic data can help with classification tasks. Mukhopadhyay et al (CDG), achieve 71.9% on imagenet 1k with fully self-supervised diffusion objective without text condition. Mukhopadhyay et al (DifFormer) achieve 76.0% in imagenet 1k, by learning an attention head to fuse the features from the diffusion model. Xiang et al (DDAE) using an unconditional diffusion model’s features yields competitive results on CIFAR10 and TinyImageNet. SODA (Drew et al), DAE (Chen et al) and DiffMAE (Chen et al) all show reasonable performance on Imagenet, learning an encoder and using a diffusion based decoder model. On the other hand DiffusionClassifier (Li et al) prompts a pre-trained diffusion model to find few shot classification by measuring the reconstruction loss. JDM (Deja et al), jointly learns a diffusion model and representations for classification. HybViT (Yang et al) shows vit can be used as generative models and shows both generative and representation features of the hybrid model.*
> >
> >
> > [1] Michael Fuest, Pingchuan Ma, Ming Gui, Johannes S. Fischer, Vincent Tao Hu, Bjorn Ommer. Diffusion Models and Representation Learning: A Survey.

---

### Official Review · Reviewer_5ayv · 2024-11-06

**Soundness:** 3
**Presentation:** 4
**Contribution:** 2
**Rating:** 5
**Confidence:** 4

**Summary:**

This paper analyzes generative pre-training on videos, in particular with an autoregressive decoder-only model. It aims to study the different components that go into these models and demonstrate that this is a useful paradigm for a strong general-purpose image and video encoder. The paper trains a Llama-style transformer model with a discrete tokenizer on 1T video tokens and shows good performance on a wide array of tasks spanning image classification, robotic manipulation and video tasks like tracking, action recognition and forecasting, while also showing the effect of different design choices in the tokenizer, model architecture, and token resolution.

**Strengths:**

1) I think this work is valuable as a systems paper that explores the benefits of various components used for visual generation, as well as their effect on creating strong video representations.
2) The paper measures downstream performance of the proposed encoder on a very large array of diverse tasks, ranging from Imagenet classification to robotic manipulation. This is very valuable and clearly demonstrates the strengths of the Toto model.
3) The appendix contains many useful details that demonstrate other strengths from the Toto model, lending credence to the thoroughness of the study.
4) The paper is extremely well written and presented. It was a pleasure to read.

**Weaknesses:**

1) There is no major technical novelty in this work as autoregressive visual generation has been done before (e.g PARTI from Google).  The introduction mentions "necesary architectural changes for scaling to videos" but it's not mentioned clearly what those are - all the components described are standard for autoregressive decoder-only visual generation. The only thing I see is the attention pooling for token weighting, but that's referenced from a prior work as well. However, the value in this paper is from the analysis of the design choices, and this is not a weakness provided the experimental analysis is thorough and insightful.

2) I think the analysis is missing some important pieces. While there are a lot of experiments, the takeaway and why is not really answered. For example, why is dVAE and patch-dVAE worse than VQGAN? Is the difference entirely due to label leak through perceptual loss? In Table 6, why is Mamba so much worse than Toto?  Furthermore, is the comparison to GPT2 really fair? Toto is based on Llama which is a much newer and upgraded architecture - shouldn't it be expected that Toto would be better? On Table 8, why is Toto much worse than other generative approaches for action recognition? In general, my criticism is that for the results section, a lot of results are presented without much analysis or experiments to explain why the results are there, which really limits the insight of the paper. I'd appreciate clarity on this if I missed something.

3) isn't the scaling behavior somewhat trivial? We know that Llama already exhibits these scaling behaviors, is it surprising that a generative decoder-only model with the same architecture but a different tokenizer would have similar scaling properties?

4) Where is the part of the paper that says/shows relative positional embeddings are better than absolute ones? this is mentioned in the introduction and conclusion but does not seem to be ablated or evaluated anywhere in the paper.

**Questions:**

I am assigning this paper a rating of 5 for now and will gladly increase my rating if my weaknesses listed are addressed and some of the questions below are answered, or if I misunderstood something and it is clarified.

1) I don't really think that VQGAN being "corrupted" by ImageNet labels makes much sense, it's the most commonly used discrete visual tokenizer and deserves careful analysis along with dVAE. If image net label leak was a concern, it would make more sense to evaluate a different image task than simple classification - maybe segmentation or object detection would make sense instead (though i understand that Imagenet linear probing is a standard measure of visual represenation quality)

2) What is the single insight of the Toto Model? It seems like a generative autoregressive decoder-only video model performs comparably with other visual encoders but it's not necessarily clearly better on all the tasks. Is there a reason this is expected?

3) Is there a reason that video-language tasks were not evaluated? Arguably the most common task with video encoders today is video + language (ie captioning, QA, etc). This would really demonstrate the strength of the encoder, and i think could provide a more compelling case for Toto being a strong encoder and for large scale video pre-training.

---

> ### Author Response · Authors · 2024-11-17
> **Official Response by Author for the Reviewer 5ayv**
>
> We thank the reviewer for comments on the detailed experiments and credence in the paper and we thank the reviewer the comment “The paper is extremely well written and presented. It was a pleasure to read”. Here we answer the questions the reviewer is asking, and we are happy to answer more questions or run more experiments.
>
> **There is no major technical novelty**
>
> As stated in the abstract, we empirically study generative pretraining on videos and this paper does not describe a novel method. Instead, it studies a straightforward, yet must-know baseline given the recent progress of large language models pre-training for self-supervised vision pretraining. As the reviewer mentioned, the paper has the design choices and all the insights for next token pre-training for vision tasks and scaling laws. With the major success from large language model pretraining, we believe this paper establishes a good experimental study of video pretraining, and explore the different design choices, that provides a path to building better models in the future. This analysis is also thoughtful from the data side, and only used publicly available data sources for the pretraining for reproducibility purposes. As mentioned, we plan to release the model weights and training code.
>
> **The takeaway and why is not really answered**
>
> We agree with the reviewers that some of the experiments do not have very conclusive insights. We did not want to make strong claims with partial observations, and some of these results are not enough to come to a strong conclusion. However, we will discuss these results here, and will add this to the paper.
>
> **1) dVAE and patch-dVAE worse than VQGAN?** In both cases, dVAE is taken at 16x16 resolution, but as shown in table 3, 32x32 dvae is better. This is the main reason behind the resolution with dvae tokens, they became blurry at low resolution, however, as shown in the table 4, we can still pre-train at 16x16, but a little bit of finetuning and change of rope base value can leads to the better performance of dvae with 32x32 resolution (which is expensive). As also shown in Revenge of the ViT.
>
> **2) Is the difference entirely due to label leak through perceptual loss?** Not necessarily, VQ-GAN training also has a discriminator loss, and also the training data is different eg Imagenet 1k for vqgan vs dalle style millions of data for dvae.
>
> **3) Why is Mamba so much worse than Toto?** During the pretraining, we found the instabilities in mamba probling performance. Also, for the experiments in the table we matched the number of parameters, not flops, this might be a reason for this behavior.
>
> **4) Furthermore, is the comparison to GPT2 really fair?** We agree that llama has more recent advances in architecture than gpt2, and that was the main point of this table, to use llama rather than gpt2, and make use of rope embeddings, silu, and rms norm. Apart from this there are not much difference in llama to gpt2.
>
> **5) Why is Toto much worse than other generative approaches for action recognition?** We agree on this paper, and part of this we had to attribute to the tokenization, a) it was chosen from imagenet experiments, b) the tokenizer is frame based, not video based, at the time of pre training, we didn't have any open source image+video tokenizers, but this could be also a reason for the drop in performance, specially, the number of frames are limited by the tokens, and if we could use a video tokenizer, we can have higher temporal resolution.
>
>
> **Isn't the scaling behavior somewhat trivial?**
>
> Although scaling is indeed a trivial property, we are the first to quantify scaling laws for vision pre-training. We wanted to emphasize that, a) we have a full recipe to scale the model training, with proper mu-parameterization with fixed learning rate, so that we don't need to train multiple big models on lots of compute to tune hyper parameters, and b) we quantify the scaling laws (L = 7.42 * C ^ {-0.0386}) and the main insight is the rate vision only models scale is slower than langage only models (at least for this tokenizer and our mixture of data and model architecture), this opens up new research directions to build and qualify the scaling on vision only models to and try to match as langage only scaling with better data mixtures, architectures and tokenizers. As for the different scaling behavior, this is mainly due to the type of tokens, visual tokens which are less compressed compared to text tokens.
>
> **relative positional embeddings are better than absolute ones?**
>
> For the rope vs absolute positional embedding we were referring to the llama vs gpt2 comparison. But it has other factors such as RMS norm and silu activations. To have a fair comparison, we will run experiments on llama with RoPE and Absolute positional embeddings. We will get these results before the end of the discussion period.

---

> > ### Author Response · Authors · 2024-11-17
> > **Response -- continued**
> >
> > **I don't really think that VQGAN being "corrupted" by ImageNet labels**
> >
> > We agree VQ-GAN is a very commonly used tokenizer (still mostly for image generation tasks.). For the purpose of this work we wanted to be fully self-supervised, without any knowledge of supervision at any point in the pipeline, as it builds a clear baseline for the next token video model. We did compare VQGAN and dVAE, as our table 3 shows both did similar on imagenet, although dvae required more tokens. We are happy to try to train VQGAN without perceptual loss and then use it for training a smaller toto model and add these comparisons, if this is something the reviewer would accept, or any other recommendations would be helpful as well.
> >
> > **What is the single insight of the Toto Model?**
> >
> > We study the behaviors of auro regressive video models for learning representations, and ablated all the design choices with minimal inductive bias, and zero supervision at any part of the training. Apart from that we have a wide suit of tasks and the comparable performance on these wide range of tasks. Finally, we also established a large scale experiment, to quantify scaling law for video next token prediction models for the first time. More specifically, we showed even with discrete tokens we can pre train at low-resolution and fine-tune at high resolution with RoPE embeddings to get better performance (Table 3), with decoder only model, we showed that best performance is happening in the middle of the model depth for all tasks and all model sizes (Fig 9), For fine tuning, we showed that if the model is pre trained with videos, it can still be finetuned with full attention without loss in performance (table 13), We also showed the benefits mu-parameterization for visual next token prediction tasks (Table 16).
> >
> > **Is there a reason that video-language tasks were not evaluated?**
> >
> > We chose to operate on the vision only domain, to minimize the scaling effects caused by language, and focus on the core visual scaling part. Many recent works suggest that language has a stronger influence than its visual counterpart, and we wanted to isolate this and carefully study the visual behavioral. We believe that involving language in this study is out-of-scope for this paper and we will consider exploring it in future work.

---

### Meta-Review · Area_Chair_YXvq · 2024-12-21

**Metareview:**

This paper proposes a simple auto-regressive style of transformer for video-pretraining, based on analysis of various model components.  The model, Toto, is tested on various tasks ranging from images, to video, to robotics and demonstrates compelling results.

The reviewers recognize several strengths in the work.  The main strength is the analysis on the various components that go into generative pre-training.  The takeaways from these analysis results can also spark further discussion in the community.

There are several quesetions and weaknesses raised by the reviewers.  They range from
* lack of related works and discussion
* qualified experimental comparisons
* poor performance which is not explained or addressed

Some of these points are addressed by the author response, but other points remain open, leaving some reviewers unconvinced and lowering their initial scores.  After extensive discussion, the reviewers have not reached a consensus.  Three of the reviewers are inclined towards rejection, with scores of 1, 3 (cPsN) and 5.  One reviewer remains positive and gives a final score of 8 (weak accept), leaning towards 6 (borderline accept).  The reviewer cPsN's score is discounted as they wrote a short review and did not participate in the discussion.

The AC has carefully read through the paper, reviews, author responses and discussion amongst reviewers.  Unfortunately, the recommendation is to reject the work.  It is a difficult decision, as the main strenght of the work would make it a great contribution to the the community - barring the current weaknesses which make it incomplete and not yet ready for publication.

One main issue is related to the overall positioning of the contributions.  If the paper is positioned as an analysis work, where its strengths lie, it lacks certain experiments to make it more comprehensive and give stronger conclusions and insights.  Yet if it is put forth as a new system, as the current writing seems to suggest, there is little technical novelty to support the design. The results are quite far from state-of-the-art, despite the authors claiming that the results are "competitive". While this is a subjective call, what "competitive" really is, there is definitely a need for analysis and discussion for why such a gap exists.

The authors are recommended to address the various comments (there are many detailed suggestions!) from the reviewers and resubmit the work at a future venue.

**Additional Comments On Reviewer Discussion:**

The reviewers discussed
(1) technical novelty of the work - to which they agreed that the paper does not have
(2) the need for state-of-the-art performance; the general consensus is that the paper does not need to have, so long as the "negative" results are explained, which it currently does not

---

### Decision · Program_Chairs · 2025-01-22

Reject